# The assembly platform FimD is required to obtain the most stable quaternary structure of type 1 pili

Dawid S. Zyla [1,6], Thomas Wiegand[2,3,4], Paul Bachmann [1], Rafal Zdanowicz [1], Christoph Giese [1], Beat H. Meier[2], Gabriel Waksman[5], Manuela K. Hospenthal [1,5] ✉ & Rudi Glockshuber [1]

Type 1 pili are important virulence factors of uropathogenic *Escherichia coli* that mediate bacterial attachment to epithelial cells in the urinary tract. The pilus rod is comprised of thousands of copies of the main structural subunit FimA and is assembled in vivo by the assembly platform FimD. Although type 1 pilus rods can self-assemble from FimA in vitro, this reaction is slower and produces structures with lower kinetic stability against denaturants compared to in vivo-assembled rods. Our study reveals that FimD-catalysed in vitro-assembled type 1 pilus rods attain a similar stability as pilus rods assembled in vivo. Employing structural, biophysical and biochemical analyses, we show that in vitro assembly reactions lacking FimD produce pilus rods with structural defects, reducing their stability against dissociation. Overall, our results indicate that FimD is not only required for the catalysis of pilus assembly, but also to control the assembly of the most stable quaternary structure.

Chaperone-usher (CU) pili are long, linear bacterial appendages, anchored to the outer membrane of gram-negative bacteria. CU pili are responsible for pathogen–host cell interactions, evasion of the host immune system, as well as biofilm formation, making them essential virulence factors[1–3]. The best-studied CU pili are type 1 and P pili, contributing to the pathogenicity of uropathogenic *Escherichia coli* (UPEC) during urinary tract infections (UTIs).

Type 1 pili are mainly produced by UPEC during infection in the bladder (cystitis)[4]. During later stages of infection when UPEC travel farther up the urinary tract to the kidneys (pyelonephritis), P pili become the main CU pili present on UPEC[5]. Type 1 and P pili share a similar architecture, with a tip fibrillum attached to a long rod anchored in the outer membrane (Fig. S1)[6–8]. The attachment of pili, and thus of bacteria to their host cells, happens via specialised adhesin proteins, FimH for type 1 pili and PapG for P pili, positioned at the distal end of the fibrillum[9,10]. Together with their connector proteins, these

adhesins constitute the tip fibrillum, which is connected to the distal end of the pilus rod (Fig. S1a). In the case of type 1 pili, FimH connects to a single copy of FimG and FimF to create a short tip fibrillum. By contrast, the P pilus tip fibrillum is longer and is made up of the connector proteins PapF, PapE (5–10 copies) and PapK. The rod is comprised of the main structural subunit, FimA in type 1 pili and PapA in P pili and is terminated by FimI in type 1 pili and PapH in P pili[11].

Pilus subunits are transported through the general SEC-pathway to the periplasm, wherein unfolded polypeptide chains are first oxidised by the periplasmic disulfide oxidoreductase DsbA, leading to the formation of a single structural disulfide bond[12,13]. All structural subunits of the pilus on their own are unstable and prone to aggregation because they consist of an incomplete immunoglobulin-like fold. They lack the final, seventh β-strand, exposing a deep hydrophobic groove on their surface[14–17]. The groove has five binding pockets, P1–P5, which in order to fold properly, have to be occupied by a donor strand

[1]Institute of Molecular Biology and Biophysics, ETH Zürich, Otto-Stern-Weg 5, 8093 Zürich, Switzerland. [2]Laboratory of Physical Chemistry, ETH Zürich, Vladimir-Prelog-Weg 1-5/10, 8093 Zürich, Switzerland. [3]Institute of Technical and Macromolecular Chemistry, RWTH Aachen University, Worringerweg 2, 52074 Aachen, Germany. [4]Max Planck Institute for Chemical Energy Conversion, Stiftstr. 34-36, 45470 Mülheim/Ruhr, Germany. [5]Institute of Structural and Molecular Biology, University College London and Birkbeck, London WC1E 7HX, UK. [6]Present address: La Jolla Institute for Immunology, 9420 Athena Cir, La Jolla, CA 92037, USA. ✉e-mail: manuela.hospenthal@mol.biol.ethz.ch

originating either from the periplasmic chaperone, FimC for type 1 pili and PapD for P pili, or by the N-terminal extension (Nte) of the following subunit, in a mechanism called donor-strand complementation (DSC)[14,16–18]. The periplasmic chaperones FimC and PapD catalyse the folding reaction of disulfide-intact pilus subunits and deliver the folded subunits to a specialised outer membrane-embedded assembly platform, FimD for type 1 pili or PapC for P pili, where pilus biogenesis occurs[12].

FimD and PapC are outer membrane proteins composed of a 24-stranded β-barrel pore, an N-terminal domain (NTD), a plug domain and two C-terminal domains (CTDs). During pilus biogenesis in vivo, chaperone-subunit complexes diffuse to the assembly platform, which coordinates and catalyses pilus formation by sequentially incorporating subunits into the growing pilus[19–23], (reviewed in ref. [6]). In the pilus rod, the subunits are linked together by a process called donor-strand exchange (DSE)[14,24–26]. The cycle of chaperone-subunit incorporation is broken with the stochastic incorporation of the pilus assembly terminator, FimI for type 1 pili and PapH for P pili, which lacks the P5 binding pocket likely making it unable to undergo DSE[11,27]; thus concluding pilus rod formation.

The pilus rod is a right-handed superhelical structure composed of ~3.2 subunits per turn with an axial rise between two consecutive subunits of 7.8 Å or 7.7 Å for the type 1 and P pilus rods, respectively (Fig. S1b)[28,29]. All subunits in the rod are connected via DSE, forming an exceptionally stable fold that is resistant to extreme temperatures, denaturants and mechanical stress[28,30–33]. The superhelical quaternary structure of the pilus results in the formation of an extensive network of interactions between subunits; in the P pilus, each subunit interacts with five preceding and five succeeding subunits[33], whereas a single subunit in the type 1 pilus creates interactions with four subunits in either direction[28,34]. The additional subunit contact in P pili is achieved via the N-terminal staple region absent in type 1 pili[33]. However, the most important and extensive interaction in the pilus rod is the stacking interface between subunits n and n + 3.

These extensive intersubunit interactions explain the biomechanical properties of CU pili in response to external forces such as urine flow, whereby the helically wound rod can uncoil by breaking the stack-to-stack interactions, while adjacent subunits remain connected through the highly stable DSE interaction[28,31–33]. The stress force is dispersed over thousands of subunits, reducing the force experienced by the adhesin[35,36]. Afterwards, the pilus rod can recoil back to its superhelical quaternary structure as demonstrated by atomic force microscopy (AFM) and optical tweezers experiments[31,32,35–43].

Previously, CU pilus rods have been assembled in vitro from purified FimC–FimA, PapD–PapA or self-complemented FimA subunits[28,33,34,44,45]. FimA is the only known subunit capable of using its own donor strand to complement its fold, resulting in a meta-stable tertiary structure with an unfolding half-life of ~10 min[46]. The rate of the in vivo assembly is approximately five orders of magnitude faster than in vitro assembly from FimA–FimC complexes[45]. While, both reactions result in rods, which appear highly similar under negative stain electron microscopy (NS-EM), pili assembled in vivo showed three to four orders of magnitude higher kinetic stability against dissociation/unfolding in the presence of guanidinium chloride (GdmCl) than those assembled in vitro[28]. The source of this stability difference is not clear, but as previously reported from AFM experiments, the type 1 pilus can refold into two distinct quaternary structures differing in their refolding forces[32].

In this work, we combine biophysical assays to measure pilus stability with structural data from cryo-electron microscopy (cryo-EM) and solid-state nuclear magnetic resonance spectroscopy (NMR) to demonstrate the importance of the pilus assembly pathway for the kinetic stability against dissociation and ordered quaternary structure of the pilus rod. Our results show that pili assembled by the bacterium

in vivo are significantly more stable against dissociation/unfolding compared to pili assembled in vitro for both type 1 and P pili. Importantly, the stability of in vitro-assembled type 1 pilus rods can be restored to near-in vivo levels by the addition of FimD to the assembly reaction. Our structural and biochemical studies indicate that, while the overall structure of the differently assembled pilus rods do not differ on the macromolecular level, in vitro-assembled rods in the absence of FimD appear to contain structural defects that lead to pilus rod destabilisation. Overall, our results point to a crucial role of FimD, not only in the catalysis of pilus assembly, but also to ensure correct subunit incorporation, therefore guiding the assembly of the most stable quaternary structure.

## Results

### In vivo-assembled type 1 pili purified from *E. coli* cells are more stable than those assembled in vitro

Type 1 pili can be produced in vivo via the CU pathway and isolated by shearing them from the bacterial cell surface, or obtained via two different in vitro assembly routes, starting from either (i) self-complemented FimA monomers or (ii) FimA–FimC complexes. Pilus samples assembled in vitro and in vivo appear identical when inspected by NS-EM and do not contain any posttranslational covalent modifications (Fig. 1a, Fig. S2). We first measured the kinetic stability of in vitro pili assembled from FimA monomers and in vivo-assembled pili against dissociation/unfolding by monitoring the decrease in ellipticity at 230 nm using circular dichroism (CD) spectroscopy at pH 2.0 in the presence of increasing concentrations of the chemical denaturant GdmCl (Fig. 1b). Both samples unfolded with uniform single exponential unfolding kinetics (Fig. S3a, b). For a quantitative comparison, we used the observed rate constants of unfolding ($k_U^{6.45M}$) at 6.45 M GdmCl (within the measured GdmCl concentration range for all samples), rather than the rate constants of unfolding in the absence of denaturant ($k_U^0$), to avoid larger errors by extrapolating over a very high denaturant concentration range (Table S1). At 6.45 M GdmCl, the $k_U$ was $3.51\cdot10^{-5}\,s^{-1}$ for in vivo pili and $2.67\cdot10^{-2}\,s^{-1}$ for in vitro pili (Fig. 1b, Table S1), suggesting that in vivo-assembled pili are ~1000-fold more stable against dissociation/unfolding than those assembled in vitro, consistent with what we showed previously[28]. The unfolding kinetics of in vivo-assembled pili displayed a characteristic kink at higher GdmCl concentrations, which can be explained by a high-energy unfolding intermediate and a GdmCl-dependent change in the rate-limiting transition states[47]. Next, we compared the stability of pili that were produced via the two different in vitro assembly routes of FimA, differing by the presence or absence of the chaperone FimC (Fig. 1b). Similar results were obtained for both samples, indicating that FimC did not affect the stability of the resulting pilus rods. Lastly, we analysed whether the pilus length influenced pilus stability by measuring the stability of both short (~400 nm) and long (>1000 nm) in vivo-assembled pili (Fig. 1a, b). Once again, we obtained similar unfolding kinetics, indicating that the observed stabilities are not length-dependent above pilus lengths of 400 nm. Importantly, when a 1:1 mixture of in vitro and in vivo-assembled pili was incubated at 7.1 M GdmCl, which is sufficient to unfold both pilus samples, the resulting unfolding trace was biphasic, revealing the distinct unfolding signatures of in vitro and in vivo-assembled pili (Fig. 1c). In contrast, a GdmCl concentration of 6.1 M was only sufficient to unfold the fraction of in vitro-assembled pili in a single monoexponential reaction. The results suggest that each pilus sample was homogeneous and did not consist of a heterogeneous mixture of different quaternary structures with different stabilities.

### The presence of FimD during in vitro assembly restores pilus stability

We hypothesised that in vivo-assembled pili reach their maximum stability due to the presence of the native assembly apparatus during

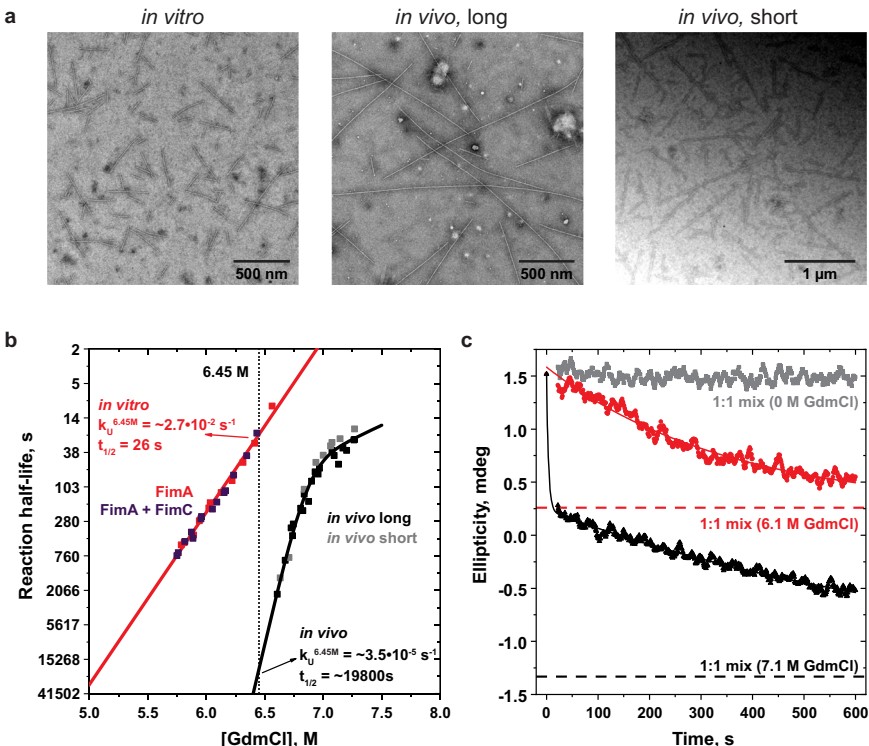

**Fig. 1 | Stability against dissociation/unfolding by GdmCl of in vitro and in vivo-assembled pili. a** Representative NS-EM micrographs of in vitro (left), long in vivo (>1000 nm) (middle) and short in vivo (~400 nm) (right) assembled pili. A scale bar is shown for each micrograph. **b** Dependence of the half-lives (logarithmic scale) of the apparent first-order unfolding/dissociation reactions on GdmCl concentration of in vitro-assembled pili from FimA monomers (red), FimA–FimC complexes (slate blue), long in vivo-assembled pili (black) and short in vivo-assembled pili (grey). Data were fitted globally with a linear function (two-state reaction, solid red line) or with a model assuming a high-energy unfolding intermediate (solid black line). The reaction half-lives (logarithmic scale) were deduced from the recorded first-order rate constants ($k_U^{obs}$). **c** Identification and quantification of different quaternary structures in mixtures of pili assembled in vivo and in vitro via GdmCl-induced unfolding. A 1:1 mixture of in vitro and in vivo-assembled pili was unfolded by 6.1 M GdmCl (red) or 7.1 M GdmCl (black) at pH 2.1, and the CD signal decrease at 230 nm was recorded. A control experiment at zero GdmCl (grey) showed no dissociation/unfolding at pH 2.1. Each pilus rod preparation alone showed uniform, single exponential unfolding behaviour. Pili assembled in vivo proved to be resistant against unfolding at 6.1 M GdmCl. At 7.1 M GdmCl, two unfolding phases are visible – a fast phase for the less stable in vitro-assembled pili and a slower one for the more stable in vivo-assembled pili. The data at 6.1 M GdmCl were fitted with a single exponential and those at 7.1 M GdmCl with a double exponential decay equation, respectively.

pilus biogenesis. In vivo, the outer membrane-embedded usher pore FimD catalyses the assembly of the tip fibrillum, containing FimH, FimG and FimF, followed by the pilus rod, containing FimA. To investigate whether FimD is responsible for the increased stability of in vivo-assembled pili, we assembled FimD-catalysed pili in vitro (hereafter denoted as FimD-cat) in a reaction where FimDCH was primed by the addition of FimC–FimF (Fig. 2a, b). We first measured the kinetic stability against dissociation/unfolding by GdmCl as before (Fig. 2c). The resulting unfolding half-lives (t½) at 6.45 M GdmCl showed that the FimD-cat pili (t½ = ~1700 s) were almost as stable as the in vivo-assembled pili (t½ = ~19,800 s) and significantly more stable than pili assembled in vitro (t½ = 26 s) at 6.45 M GdmCl (Table S1). Moreover, the dependence of the unfolding half-life on GdmCl of FimD-cat pili also displayed the same characteristic kink at high GdmCl concentrations as that of in vivo-assembled pili (>6.75 M). All unfolding traces showed uniform single exponential unfolding behaviour as before (Fig. S3c). To rule out that the assembly reaction temperature or the presence of detergent (required to solubilise FimD in the reconstituted FimD-cat assembly reaction) influenced the resulting pilus stability against GdmCl, we assembled pili in the presence or absence of FimD at 25 and 37 °C, and measured the stability of in vitro-assembled pili in the presence of 0.05% n-dodecyl-β-D-maltopyranoside (DDM). The results showed that neither the temperature during assembly nor the presence of DDM affected the stability of the assembled pili (Fig. S4). Moreover, priming of the usher with FimC–FimG, instead of FimC–FimF also did not affect stability (Fig. S4). Last, we performed

thermal stability measurements of our samples by monitoring the change in the CD signal at 230 nm with increasing temperature in the presence of 7.5 M GdmCl at pH 7.0, since pilus rods do not unfold at pH 7.0 below 100 °C under non-denaturing conditions (Fig. 2d). The melting temperature of in vitro-assembled pili was significantly lower (59 °C) than for pilus samples assembled in vivo (79 °C) or in the presence of FimD (78 °C). Our results indicate that in vivo-assembled pili are significantly more stable than their in vitro-assembled counterparts. Given that FimDCH needs to be primed with either FimC–FimF of FimC–FimG in our FimD-cat assembly reaction, the presence of these additional subunits cannot be avoided in FimD-cat pili, and their effect cannot be fully excluded[11]. However, our assays measure chemical or thermal stability against dissociation/unfolding, with subunit dissociation likely being the rate-limiting step. Given that GdmCl- or temperature-induced dissociation is not initiated from the pilus ends, but rather affects all subunits simultaneously and equally, we do not think that the presence of the tip fibrillum explains the observed differences. In addition, the tip fibrillum subunits constitute less than ~5% of all pilus subunits in our FimD-cat sample, making their dissociation/unfolding invisible in our CD assay. Moreover, the fact that FimD-cat pili do not differ in stability when the last subunit of the tip fibrillum is FimG or FimF (Fig. S4) speaks against an influence of the tip fibrillum on pilus rod stability. Therefore, a more likely explanation is that FimD plays a crucial role in the formation of the most stable quaternary structure because it sequentially adds subunits to the growing pilus rod.

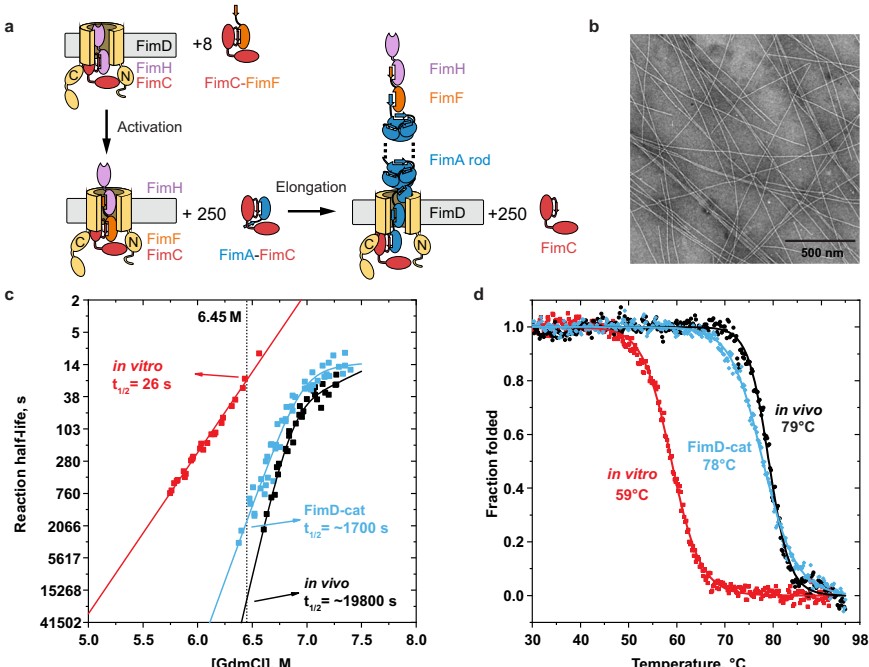

**Fig. 2 | FimD restores the stability of in vitro-assembled pili. a** Schematic diagram of FimD-catalysed (FimD-cat) pilus assembly in vitro. The FimDCH complex was activated with eight-molar excess of FimC–FimF, followed by the addition of a 250-fold molar excess of FimC–FimA. The various pilus subunits are labelled and coloured accordingly. The N- and C-terminal domains of FimD are also indicated. **b** Representative NS-EM micrograph of FimD-cat pili. The scale bar is shown in the figure. In total, 60 micrographs were analysed. **c** Dependence of unfolding half-life on GdmCl concentration for in vitro-assembled pili (red), FimD-cat pili (blue) and in vivo-assembled pili (black). The cross-section at 6.45 M GdmCl of the V-plot shows the relative half-lives of the dissociation/unfolding reaction for each sample. All measurements of FimD-cat pili (from different assembly reactions and at different temperatures) are shown in Fig. S4. **d** CD melting curves of in vitro-assembled pili (red), FimD-cat pili (blue) and in vivo-assembled pili (black). The apparent melting temperatures are indicated next to each curve. Data were fitted according to a single two-state transition equation (solid curves) and normalised.

## Assembly-dependent stability differences are also observed for the closely related P pili

We wondered whether similar stability differences caused by the assembly pathway (in vitro vs. in vivo) are also present in other CU pili. To investigate this, we analysed the unfolding kinetics of in vitro and in vivo-assembled P pili, which are closely related to type 1 pili. We obtained an in vivo-assembled P pilus sample by shearing P pili from the *E. coli* cell surface, and an in vitro-assembled P pilus sample assembled from PapA–PapD complexes (Fig. 3a), and measured the dissociation/unfolding kinetics of both samples in the presence of increasing concentrations of GdmCl by monitoring the change in CD signal at 222 nm (Fig. 3b). Indeed, the results showed that P pili assembled in vivo are also more stable than those assembled in vitro, as we observed for type 1 pili. However, the magnitude of the kinetic stability difference at 7.2 M GdmCl was only ~10-fold ($t_{1/2}$ 125 s vs. ~1090 s, in vitro vs. in vivo, Table S1), in contrast to the ~1000-fold difference observed for type 1 pili. Notably, P pili showed even higher stability against dissociation/unfolding at high GdmCl concentrations than type 1 pili. In contrast to type 1 pili assembled in vivo, the dependence of the logarithm of $k_U^{obs}$ on GdmCl did not deviate from linearity at high GdmCl concentrations for P pili assembled in vivo (Fig. 3b). Our findings suggest that P pili can also only reach the most stable quaternary structure when the assembly is catalysed by their assembly platform (PapC) and that this may be a general feature of CU pili.

## Cryo-EM structures of pilus rods obtained by different assembly pathways are highly similar

To investigate whether the stability differences observed in the GdmCl-dependent dissociation/unfolding experiments are linked to the quaternary structure of pili, we determined the cryo-EM structures of our pilus rod samples (Fig. 4a, Fig. S5). Previously determined structures of in vivo-assembled pilus rods (Fig. S6), resolved at ~4.2 Å[28,29], did not allow us to identify potentially subtle differences in structure, such as specific side-chain orientations. Therefore, we set out to determine another structure of in vivo-assembled pili at a higher resolution. Our structures of in vivo-assembled, in vitro-assembled and FimD-cat pilus rods were resolved to 2.85, 2.69 and 2.52 Å, respectively (Fig. 4a, Fig. S5). As with previous CU pilus rod structures determined by cryo-EM, the parts of the structure facing the lumen are better resolved than the more flexible loop regions around the pilus exterior (Figs. S5a–c and S6)[32,33,46]. We modelled six adjacent FimA subunits linked through donor-strand interactions and refined the models against their respective cryo-EM maps (Fig. 4b). The Cα root mean square deviation (RMSD) from global alignment of the three models was less than 1 Å in each case (0.68 Å, 0.69 Å and 0.25 Å for FimD-cat vs. in vivo, FimD-cat vs. in vitro and in vitro vs. in vivo, respectively) implying that all three structures are highly similar. Moreover, the helical parameters of twist and rise were also similar in all type 1 pilus rod structures (Table S2). More detailed analyses did not show any significant differences in residue contacts or subunit–subunit interfaces between all three models (Fig. S7). Reasons for this could be that the modest and heterogeneous resolutions do not allow for precise determination of side-chain conformations/rotamers and/or that helical reconstruction results in averaging of differences that are not present in every pilus subunit. In summary, our cryo-EM structures did not reveal the source of the observed stability differences between the pilus samples, so that other methods were required to pinpoint these potentially more subtle differences.

## Solid-state NMR spectra for in vivo- and in vitro-assembled pili reveal spectral differences

NMR chemical-shift values are known to be highly sensitive to small conformational changes and allow for a site-specific investigation of

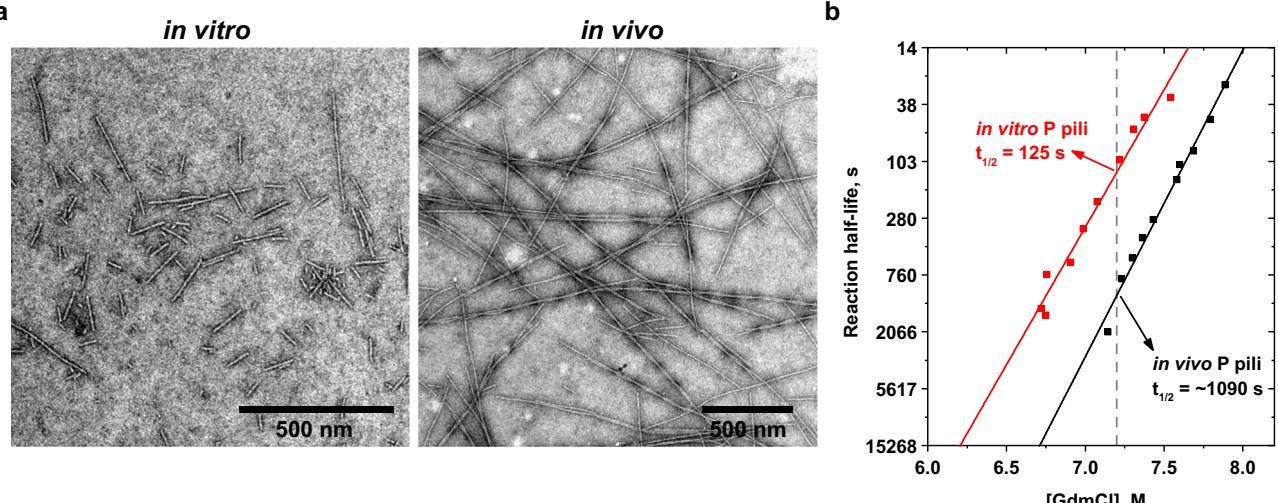

**Fig. 3 | In vitro-assembled P pili also display lower stability compared to in vivo-assembled P pili. a** Representative NS-EM micrographs of in vitro (left) and in vivo (right) assembled P pili. A scale bar is shown on each micrograph. In total, 50 micrographs were analysed for each sample. **b** Kinetic stability measurements, comparing the stability of in vitro and in vivo-assembled P pili against dissociation/unfolding by GdmCl.

such differences. We thus turned to solid-state NMR to assess possible differences between the in vitro- and in vivo-assembled pilus rods. Given that every FimA subunit in the pilus will contribute to the NMR signal, we anticipated four possible scenarios in these experiments. First, all peak positions and intensities would be identical if no structural differences were present between in vitro- and in vivo-assembled samples. Second, if structural differences were present between the two samples, and each sample was homogenous, the affected residues would be expected to have different chemical shifts and thus produce spectra with different peak positions. Third, peak splitting may occur if more than one population of FimA conformations existed within a sample. The intensity and number of split peaks (for each residue) would depend on the relative proportion of FimA subunits adopting each conformation. Lastly, broadening of resonances in one sample compared to another may occur if a fraction of FimA subunits adopted a heterogeneous distribution of conformations at those residues, for example due to packing errors.

We produced $^{13}$C, $^{15}$N-labelled in vitro- and in vivo-assembled pilus samples and recorded 2D $^{13}$C–$^{13}$C Dipolar Assisted Rotational Resonance (DARR)[48,49] and 2D $^{15}$N–$^{13}$C NCA spectra. For our analyses, we made use of previously published sequential resonance assignments for type 1 pilus rods, obtained in a similar manner from FimC–FimA complexes[34]. In contrast to this previous study, our sample was assembled in the absence of FimC[44]. Figures S8 and S9 show the corresponding highly resolved spectra indicating homogeneous sample preparations. Interestingly, the spectra were not identical and the resolution was best for the in vivo-assembled sample. Minor chemical-shift differences were observed for a subset of resonances (Table S3 and Fig. S10). Additionally, we observed peak doubling and/or line broadening for a subset of FimA residues (V17, V23, V37, A69, A72, A93, I103, T128, F154 and Q159) in the in vitro sample (Table S3). This became particularly apparent in the alanine region (Fig. 5), as well as in the NCA spectrum for residues V23 and T128, for which clearly visible peak splitting was detected (Fig. 5). Such effects were not observed for the in vivo pili. We also observed an additional threonine spin system appearing in the in vitro pili at high $^{13}$C$\beta$ shift in the indirect dimension (73.7 ppm), possibly also belonging to the subset of FimA subunits exhibiting small conformational differences in the in vitro sample. The intensity ratios of the two resonances for those residues revealing evident peak splitting was ~2:1, although the quantification was complicated by the cross-polarisation steps used (Table S4). To test

whether our observations of the in vitro pili were reproducible, we recorded solid-state NMR spectra on a second, independently prepared sample, showing that the spectra were indeed nearly identical (Figs. S11–S13). We thus conclude that the spectral differences between in vitro- and in vivo-assembled pilus rods point to structural differences associated with the different assembly schemes.

## The presence of FimD during the assembly of in vitro pilus rods produces the same conformation observed for the in vivo pili

We next wondered whether the conformational differences observed between the NMR spectra of the in vitro- and in vivo-assembled pilus rods might be related to the different dissociation/unfolding kinetics observed. For that purpose, we produced $^{13}$C, $^{15}$N-labelled in vitro pilus rods assembled in the presence of FimD, whose unfolding kinetics resembled those of the in vivo pili, and were thus distinct compared to the in vitro pilus rods assembled in the absence of FimD. Figure S8c shows the aliphatic region of the $^{13}$C–$^{13}$C DARR spectrum and comparison with Fig. S8b revealed that the spectrum is identical to the one of the in vivo pili (for the NCA spectra see Fig. S9). This was further corroborated by the spectral fingerprints (shown in Fig. 5) which revealed that the spectral differences observed in the in vitro pilus rods assembled in the absence of FimD can no longer be detected in the pili assembled in vitro in the presence of FimD (FimD-cat).

## Structural defects in in vitro-assembled pili explain their reduced stability

We reasoned that the spectral differences observed between the in vitro- and in vivo-assembled or FimD-cat pili could arise due to defects introduced into the quaternary structure of in vitro-assembled rods. Such defects could stochastically be introduced every now and then as FimA subunits are incorporated one-at-a-time in the absence of the usher, or alternatively, they could arise as a result of longer rods being assembled *en bloc* from previously assembled shorter rod fragments. Regardless of how such defects are generated, the sites at which they occur would represent weak spots in the pilus structure, which in the presence of a chemical denaturant like GdmCl, could become destabilised and lead to rod fragmentation triggering complete dissociation and unfolding. We used analytical size exclusion chromatography (SEC) to more precisely study the molecular species produced after incubating in vitro- or in vivo-assembled type 1 pili with increasing concentrations of GdmCl

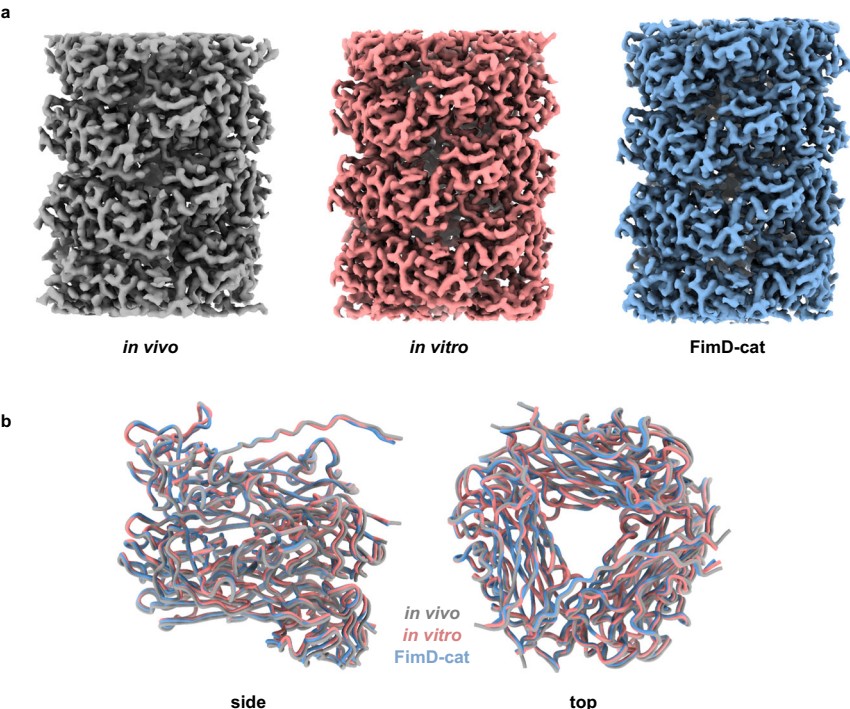

**a**

*in vivo*          *in vitro*          **FimD-cat**

**b**

*in vivo*
*in vitro*
FimD-cat

side                    top

**Fig. 4 | Cryo-EM structures of pili assembled in vivo, in vitro and in vitro in the presence of FimD (FimD-cat). a** Cryo-EM maps obtained for pili formed by different assembly pathways: in vivo-assembled (grey), in vitro-assembled (red) and FimD-cat (blue). **b** Superposition of all three models shown in a side and top view.

(Fig. 6a, b, Fig. S14). In vivo-assembled pili, without prior incubation with GdmCl or after incubation with 4.1 M GdmCl, showed a single peak in the void volume, corresponding to molecular species larger than ~5 MDa (longer than ~250 nm). This profile suggested that the pilus rods remained intact throughout this experiment. After incubation with 7.5 M GdmCl, the in vivo-assembled pili showed complete dissociation and subsequent unfolding of the FimA monomer (Fig. 6a). By contrast, pili assembled in vitro, without prior incubation with GdmCl, produced SEC profiles consisting of a void volume peak, a broad shoulder containing smaller pilus fragments of different lengths, and a peak of the unfolded FimA monomer. Following incubation with 4.1 M GdmCl, the smaller fragments became even more abundant, and after incubation with 6 M GdmCl, the pili had almost fully dissociated and unfolded (Fig. 6b). This suggested that FimA unfolding is preceded by pilus rod fragmentation under these conditions. We then asked whether the GdmCl-induced pilus fragments can be reassembled *en bloc* into longer pili (Fig. 6c). To do this, we incubated the in vitro-assembled pili with 4.5 M GdmCl to achieve rapid dissociation to smaller fragments, and immediately diluted this sample with non-denaturing buffer (20 mM sodium acetate [pH 5.0], 150 mM NaCl) to halt further fragmentation. Then, the fragments were concentrated to a FimA monomer concentration of 200 μM and further incubated at 37 °C for 16 h to reassemble the short rod fragments into longer rods. The reaction yielded a single void volume peak in the SEC experiment, demonstrating that the fragments indeed reassembled quantitatively into longer pilus rods (Fig. 6c). This was confirmed with NS-EM of the reassembled rods (Fig. 6d). In addition, these *en bloc*-reassembled in vitro pili displayed the same reduced stability as those in the original in vitro pilus preparation from which they were obtained (Fig. S15). Taken together, these results suggest that when pili were assembled in vitro from FimA monomers or FimA–FimC complexes in the absence of FimD, structural defects occurred in the resulting pilus rods, likely at the positions where small fragments associated to longer fragments. When such in vitro-assembled pilus rod samples are subjected to medium to high GdmCl concentrations, rapid fragmentation occurs at the sites of quaternary structural defects, followed by the rapid dissociation/unfolding of the resulting small fragments. By contrast, this does not occur for the much more stable in vivo-assembled pilus rods that do not contain structural defects, nor show length-dependent stability differences (Fig. 1a, b). Therefore, this provides the most plausible explanation for the ~1000-fold lower kinetic stability of in vitro-assembled pilus rods.

## Discussion

CU pili are virulence factors displayed on the surface of bacterial cells with important roles in host-cell interaction, evasion of the host immune system and biofilm formation[1–3]. Two archetypal CU pili are the type 1 and P pili from UPEC, which allow for bacterial attachment to host cell glycoproteins in the bladder and the kidney, respectively[4–6]. CU pili are assembled in vivo by the bacterial cell using the chaperone-usher pathway, but they can also be assembled in vitro at a slower rate, from purified FimC–FimA, self-complemented FimA subunits (type 1 pili) or PapD–PapA (P pili)[23,28,33,34,44–46]. While the resulting rod structures appear highly similar when visualised by NS-EM, in vivo-assembled type 1 pili display three to four orders of magnitude higher kinetic stability against GdmCl-induced dissociation/unfolding[28]. Here, we set out to understand the underlying mechanism responsible for this marked and unexpected difference in stability.

When we compared the kinetic stability of in vitro- and in vivo-assembled type 1 pili, we observed a pronounced stability difference, irrespective of pilus length, the presence of the chaperone FimC or different buffer conditions in the assembly reaction, and whether stability was tested by GdmCl-dependent dissociation/unfolding kinetics or thermal unfolding (Figs. 1b, 2c, d, Fig. S4). Strikingly, when in vitro pili were assembled in the presence of the activated usher FimD (Fig. 2a), the stability was restored to levels similar to those observed for in vivo-assembled pili (Fig. 2c, d). Furthermore, this phenomenon of a stability discrepancy does not appear to be unique to type 1 pili, as the closely related P pili displayed analogous stability differences

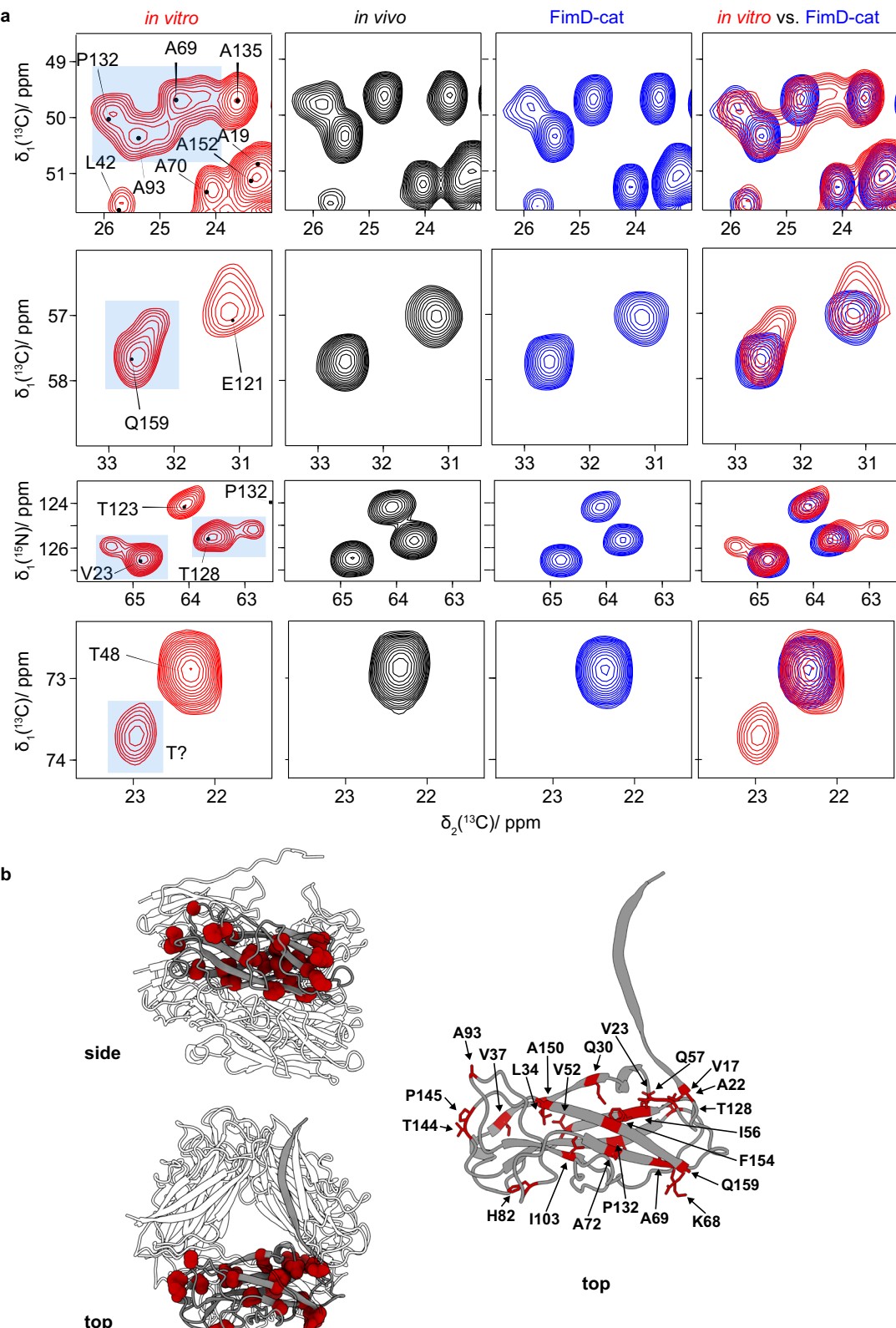

**Fig. 5 | Solid-state NMR spectra of in vitro-assembled pili differ from those of the in vivo-assembled and FimD-cat pili. a** Spectral fingerprints of 2D DARR spectra of in vitro-assembled pili (red), in vivo-assembled pili (black) and FimD-cat pili (blue). The in vitro-assembled pilus rod sample produces peak doubling and/or line broadening compared to the in vivo-assembled and FimD-cat samples, which are highly similar. Resonance assignments were taken from the BMRB database, accession code 25334[34]. **b** Mapping of residues that produced peak doubling and/or line broadening (red) onto FimA in the context of the quaternary structure (left) and the isolated FimA monomer (right), where all residues are labelled.

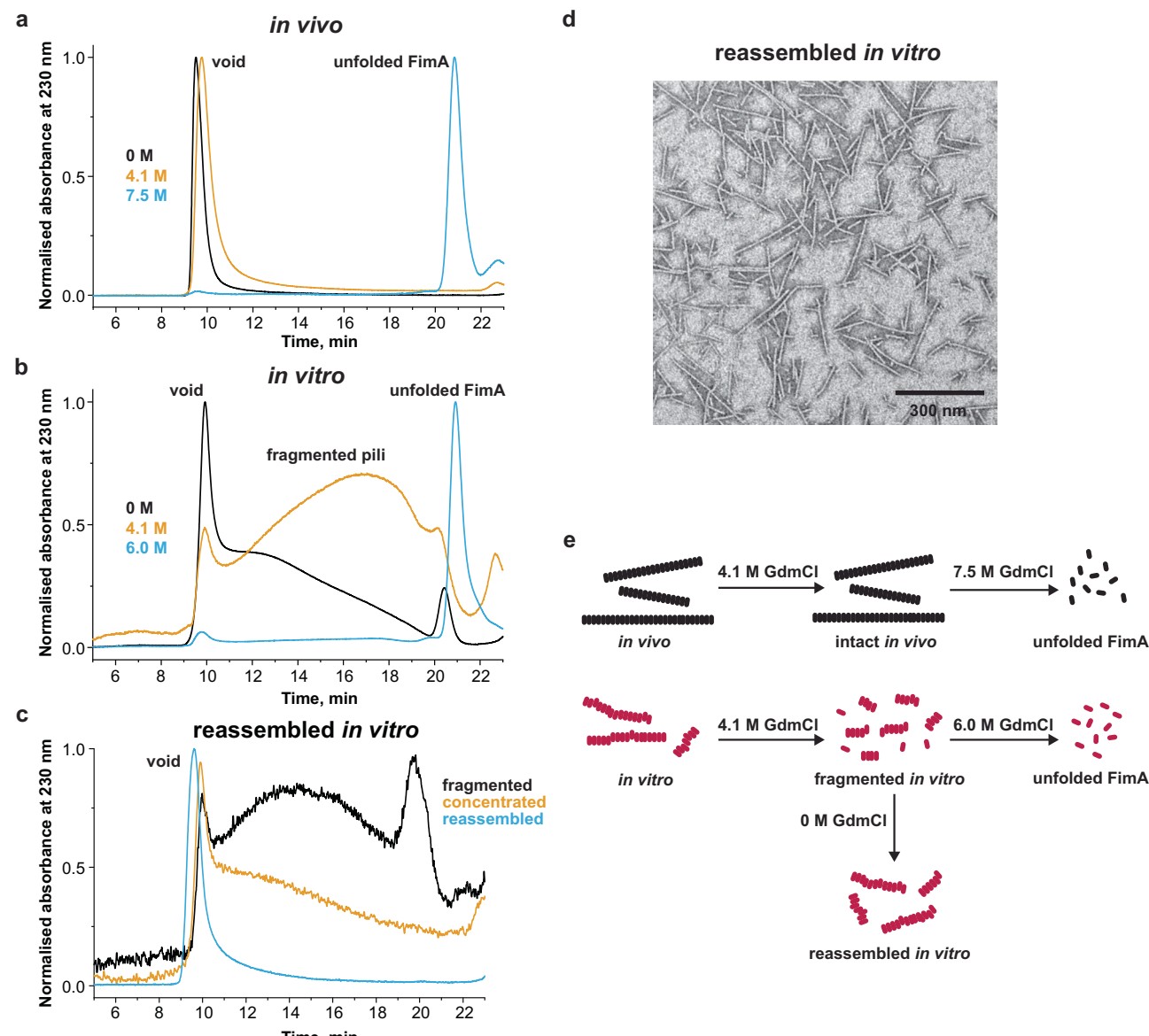

**Fig. 6 | SEC analysis of fragmentation and spontaneous reassembly of in vitro-assembled pili.** SEC profiles from a *Superose 6 10/300 column* of (**a**) in vivo and (**b**) in vitro-assembled pili after incubation at different GdmCl concentrations. The elution peaks of pilus rods longer than ~250 nm (~9.5 min, void volume), of fragmented pili (10–20 min) and unfolded FimA (~21 min) are clearly visible. **c** Reassembly of in vitro pili after fragmentation with 4.5 M GdmCl. Fragmented pili (black) were concentrated (yellow) and allowed to reassemble into longer pilus rods (blue). **d** Representative NS-EM micrograph of in vitro pili reassembled after fragmentation. In total, 65 micrographs were analysed. **e** Reaction schemes for the results shown in **a**–**d**.

depending on the assembly route (Fig. 3). Although, the in vivo-assembled and FimD-cat pili contain a tip fibrillum, its presence at one end is unlikely to cause the observed stability difference in our assay, which is a measure of global chemical or thermal stability against dissociation/unfolding. This view is supported by our finding that the stability of FimD-cat pili is independent of whether the most proximal subunit of the fibrillum is FimG or FimF. The in vivo assembly mechanism of CU pili does not allow spontaneous rod fragment assembly because the distal end of the growing rod is blocked by the tip fibrillum, and the proximal end is on the periplasmic side of the outer membrane. CU pilus assembly in vivo thus enforces pilus growth by linear and processive incorporation of FimA subunits in the specific environment of the assembly platform FimD as the only mechanism of FimA polymerisation, preventing alternative elongation mechanisms that could lead to quaternary structural defects. At present, it is not yet clear whether the usher also ensures correct positioning of FimA

subunits as they emerge from the usher pore, or perhaps limits the relative conformations/angles of each newly emerging FimA subunit with respect to previously incorporated FimA subunits, to ensure the efficient formation of a seamless quaternary rod structure.

In our experiments, P pili seemed to be more stable against GdmCl than type 1 pili (Fig. 3b), which is in contrast to force spectroscopy studies that showed that type 1 pili require a higher force for rod unwinding[31,32]. However, we think this can be explained by the nature of the two different experimental approaches. Whereas the plateau phase of force spectroscopy experiments mainly monitors the sequential unstacking of pilus subunits (governed predominantly by the strength of the n and n + 3 interfaces), our assays are a more global measure of stability against dissociation and are affected by several distinct events along the path starting from an intact pilus rod and culminating in the unfolding of all individual subunits. These distinct events include subunit unstacking and dissociation of all other subunit

interfaces in the rod, the breaking apart of donor-strand-complemented subunits and the eventual unfolding of the subunits, which is what is ultimately detected by a CD signal change at a chosen wavelength. Therefore, the key difference is that the plateau phase during a force spectroscopy experiment does not lead to subunit dissociation, yet this step is crucial in our experiments. With this in mind, our results suggest that P pili are more stable overall, while type 1 pili may be more resistant against rod unwinding, perhaps due to the difference in the buried surface area and binding energy of the stacking interface[28]. A plausible explanation for this would be that the total binding energy between a single subunit to all other subunits within the pilus is higher in P pili compared to type 1 pili[50], which might largely be due to additional structural features such as the staple region present in P pili[33]. Another noteworthy distinction was the magnitude of the stability difference observed between in vitro- and in vivo-assembled pili (~10-fold for P pili vs. ~1000-fold for type 1 pili) (Figs. 2c, 3b).

In summary, while previous experiments using AFM have primarily measured the resistance of pili against force-induced unwinding, our studies concentrated on stability against denaturant-induced dissociation/unfolding, which are mechanistically different. Consequently, a direct correlation between these two phenomena is not inherently implied. Future research is required to explore the relationship between these two distinct phenomena to elucidate the interdependence between pili's chemical and mechanical stability.

We hypothesised that the increased stability of in vivo-assembled type 1 pili compared to in vitro-assembled pili must be due to chemical changes or structural changes present only when a pilus rod is assembled in vivo or in the presence of FimD (in vitro). However, mass spectrometric analysis of in vitro- and in vivo-assembled type 1 pilus samples showed no difference (Fig. S2), ruling out covalent modifications such as isopeptide bonds[51–54]. In addition, this allowed us to rule out that the low stability of in vitro-assembled pili resulted from some FimA subunits not having formed the structural disulfide bond. In order to address structural differences, we sought to compare structures of in vivo- and in vitro-assembled pilus rods. While a number of different CU pilus rod structures were available[28,29,33,34], these did not allow for a precise enough comparison between in vitro- and in vivo-assembled samples. Therefore, we determined three high-resolution cryo-EM structures of in vitro-assembled, in vivo-assembled and FimD-cat pilus rods (Fig. 4, Fig. S5). However, after extensive analysis we were unable to pinpoint any structural differences that could explain the stability discrepancy (Fig. S7). Because the helical reconstruction method produces an averaging effect, which would mask structural defects not uniformly present in every subunit, we turned to solid-state NMR to allow us to discern whether all FimA subunits experience an identical chemical environment or not. Indeed, the solid-state NMR spectra revealed structural differences between in vivo- and in vitro-assembled samples, where the resonance positions of a few residues exhibited small shifts, peak splitting and/or line broadening (Fig. 5, Figs. S8–S10). Our experiments suggest that these structural differences occurred in approximately every third FimA subunit of in vitro-assembled pili (Table S4), and that this could explain the observed stability difference. This is further corroborated by the fact that FimD-cat pili, which display similar overall stability compared to in vivo-assembled pili, also produce identical NMR spectra. Overall, our NMR results are consistent with two possible structural models for in vitro-assembled type 1 pilus rods: (i) that each rod contains structural defects arising from the presence of at least two FimA conformations (heterogeneous in FimA conformations), or (ii) that there are two slightly different but homogenous populations of pilus rods in the sample. While such a structural polymorphism suggested by scenario (ii) has indeed been observed for FimA[55] and other pili (CS1 pilus[56]), we believe that the reproducibility of the intensity ratios of the two FimA populations from different preparations (Figs. S11–S13, Table S4), as

well as the monoexponential unfolding kinetics of in vitro-assembled pili (Fig. S3), more likely point towards structural defects occurring at short intervals along each pilus rod. Such defects would produce weak spots in the pilus rod, which could become destabilised in the presence of GdmCl and lead to rod fragmentation before undergoing complete dissociation and unfolding. Indeed, our analytical SEC experiments suggest that only in vitro-assembled pilus rods undergo fragmentation prior to complete dissociation and unfolding, and that this occurs rapidly following the addition of GdmCl (Fig. 6a, b, Fig. S14). These defects may be introduced stochastically as FimA subunits assemble one-at-a-time in the absence of FimD, or they could arise when shorter rod sections are joined together *en bloc* to become larger rods. Given that fragmented pili were able to reassemble into longer pili (Fig. 6c), which display the same stability as the in vitro-assembled samples prior to GdmCl-induced fragmentation (Fig. S15), it seems plausible that this could indeed be the mechanism with which such structural defects are introduced when pilus rods are assembled in vitro.

It has previously been observed that the secretion of subunits through usher's lumen follows a defined and continuous low-energy pathway, whereby the subunit is moved upwards and is rotated in an anticlockwise manner[20]. It is conceivable that the lumen of the usher does not only provide an energy path that drives the handover of the nascent pilus from usher's NTD to its CTD and subunit secretion, but it might also play an important role in positioning each subunit with respect to its neighbours to ensure the formation of proper residue to residue contacts and subsequent winding of the subunits into the pilus rod's most stable superhelical structure. Similar to our discovery, previous studies on the archaic Csu pilus also suggested the effect of the usher on preserving intersubunit binding energy through inhibiting premature pilin interactions, allowing the subunits to rapidly adopt their final quaternary architecture after crossing the outer membrane[57].

In summary, we showed that in vitro-assembled type 1 and P pili are less stable than those produced by the bacterial cell in vivo, and that P pili are more stable overall than type 1 pili. In addition, we showed that the reduced stability could be rescued by performing the in vitro pilus assembly reactions in the presence of the usher FimD. Our structural and biochemical data suggest that structural defects within the rods are the source of the lowered stability of in vitro-assembled samples and that these defects may be introduced when pilus rods are assembled *en bloc* from shorter fragments. These findings highlight the essential nature of the assembly platform (FimD for type 1 pili, PapC for P pili) to attain the most stable quaternary pilus rod structure. Therefore, the carefully choreographed nature of subunit binding, incorporation and secretion by the usher ensures that pilus subunits are assembled into an ordered and structurally homogeneous protein polymer free of defects to ensure maximum stability against dissociation.

## Methods

### Molecular biology

The pPap5-AraC plasmid for the arabinose-inducible production of P pili was produced by inserting the AraC promoter region upstream of the *pap* operon and replacing the native promoter region of the pPap5 plasmid[58]. Analogously, type 1 pili were produced from a plasmid (Fim-Ara) harbouring an arabinose-inducible *fim* operon as described in ref. 59.

### Production of in vivo-assembled pili

**Short pili.** For type 1 and P pilus production, *E. coli* HB101 cells (Promega) were transformed with the Fim-Ara or pPap5-AraC plasmid, respectively. In each case, a culture of $6 \times 1.5$ L of cells was grown at 37 °C to an optical density at 600 nm ($OD_{600}$) of 0.8 in 2YT medium supplemented with ampicillin (100 µg/mL), pilus expression was

induced using 0.1% arabinose and the cultures were further incubated for 4 h (P pili) or overnight (type 1 pili) at 37 °C. Cells were harvested at 6000 × *g* with a Sorvall F9S rotor (Thermo Fisher Scientific) and resuspended in 150 mL of 50 mM Tris-HCl [pH 8.0], 250 mM NaCl. Pili were sheared off the cell surface using a cell homogeniser (Ultra-TURRAX) at 24,000 rpm on ice with 1 min on and 3 min off cycles. The cell debris was pelleted by centrifugation with an SS34 rotor at 48,000 × *g* (Thermo Fisher Scientific) for 15 min at 4 °C. Supernatants were pooled, and MgCl$_2$ was added to a final concentration of 0.1 M. The mixture was incubated on ice for 1 h, until a white precipitate (pili) formed. Precipitated pili were pelleted by centrifugation with an SS34 rotor (Thermo Fisher Scientific) at 48,000 × *g* for 30 min at 4 °C and resuspended in 20 mL of ddH$_2$O. The insoluble material was pelleted by centrifugation with an SS34 rotor (Thermo Fisher Scientific) for 30 min at 48,000 × *g* at 4 °C. Pili were washed four times by ultra-centrifugation with the MLA-80 rotor (Beckman Coulter) at 200,000 × *g* for 1 h before resuspending in 250 μL ddH$_2$O.

**Long pili.** Long in vivo-assembled type 1 pili were produced as previously described in ref. 28. Briefly, a 6 × 1.5 L culture of *E. coli* W3110Δ*fimA* cells, transformed with the pCG-AC plasmid, was grown at 37 °C until the OD$_{600}$ reached 1.0, at which point pilus expression was induced by the addition of anhydrotetracycline to a final concentration of 9 ng/mL. The culture was incubated for an additional 4 h. The cells were harvested and the pili were purified as described above.

### Production of in vitro-assembled pili

**Purification of the FimA–FimC complex.** FimA, FimC and FimC$_{his}$ proteins were purified as previously described in ref. 46 and ref. 18. FimC or FimC$_{his}$ was produced in *E. coli* BL21(DE3) cells transformed with pFimC-cyt or pFimC$_{his}$-cyt plasmid under the control of the lac promoter. A culture of 6 × 1.5 L of 2YT medium supplemented with ampicillin (100 μg/mL) was grown at 37 °C until the OD$_{600}$ reached 0.6, induced with 1 mM IPTG and incubated at 37 °C for an additional 5 h. The cells were lysed in 100 mM MES–NaOH [pH 6.0] and passed five times through a pressurised cell disruptor at 14,000 PSI. The clarified lysate was dialysed against 50 mM MES–NaOH [pH 6.0] at 4 °C. The sample was applied to an SP-Sepharose column (Cytiva) and eluted using a linear pH gradient (pH 6.0–10.0) produced by applying 50 mM CAPS–NaOH [pH 10.0] to the column. Fractions containing FimC were pooled and dialysed against 50 mM MES–NaOH [pH 6.0] and reloaded on the SP-Sepharose column equilibrated with 50 mM MES–NaOH [pH 6.0]. Bound protein was eluted by applying a linear salt gradient (0–400 mM NaCl). The protein was further purified by size exclusion chromatography (SEC) using a Superdex 75 26/60 Hi-Load column (Cytiva) equilibrated with 20 mM Tris-HCl [pH 8.0], 50 mM NaCl.

The FimA–FimC complex was formed by rapid dilution of unfolded, disulfide-intact FimA in 50 mM Tris-HCl [pH 8.0], 3 M GdmCl to FimC in 20 mM Tris-HCl [pH 8.0], 150 mM NaCl. The complex was formed for 15 min at 23 °C and later applied to a Sephadex G25 column (Cytiva) equilibrated with 20 mM MES–NaOH [pH 6.0]. Protein containing fractions were pooled and loaded onto a Source 30S column (Cytiva) in the same buffer. The FimA–FimC complex was eluted from the column with a linear NaCl gradient (0–400 mM) and further purified by SEC using a Superdex 75 26/60 Hi-Load column equilibrated with 20 mM Tris-HCl [pH 8.0], 50 mM NaCl.

**Purification of the PapA–PapD complex.** *E. coli* HM125 cells were transformed with PapDA (pTRC99a)[60] encoding PapA and PapD$_{his}$, preceded by their native periplasmic signal sequences, under the control of the lac promoter. A culture of 6 × 1.5 L supplemented with ampicillin (100 μg/mL) was grown at 37 °C until the OD$_{600}$ reached 0.6, induced using 1 mM IPTG and further incubated overnight at 20 °C. Periplasmic extraction was performed by vigorously shaking (150 rpm) cells resuspended in 50 mM Tris-HCl [pH 8.0], 150 mM NaCl, 1 mg/mL polymyxin B sulfate for 2 h at 4 °C. The cell debris was pelleted by centrifugation with an SS34 rotor (Thermo Fisher Scientific) at 48,000 × *g* for 30 min at 4 °C. The supernatant was collected and applied to a 5 mL HisTrap column (Cytiva) equilibrated with 50 mM Tris-HCl [pH 8.0], 150 mM NaCl and the complex was eluted with a linear imidazole gradient (0–400 mM imidazole). Fractions containing the PapD–PapA complex were pooled, dialysed against 20 mM MES–NaOH [pH 5.5] and applied to a Source 30S column to separate the free PapD from the PapA–PapD complex by applying a linear salt gradient (0–400 mM NaCl). Fractions corresponding to the complex were pooled, concentrated and purified by SEC using a Superdex 75 26/60 Hi-Load column equilibrated with 20 mM Tris-HCl [pH 8.0], 50 mM NaCl.

### In vitro assembly of type 1 and P pilus rods

In vitro assembly of type 1 pili from self-complemented FimA was carried out as described previously[28,44]. For in vitro-assembled pili from FimA–FimC complexes, 50 μM FimA–FimC or FimA–FimC$_{his}$ complexes were incubated at 37 °C in 20 mM Tris [pH 8.0], 50 mM NaCl for 14 days. In vitro assembly of P pili was achieved by incubating 50 μM PapA–PapD$_{his}$ complex in 20 mM Tris-HCl [pH 8.0], 50 mM NaCl at 37 °C for 7 days. All pilus rod samples were purified and washed as described above.

### FimD-catalysed type 1 pilus rod assembly in vitro

The FimD–FimC–FimH (FimDCH) complex was expressed and purified following previously established protocols[11,21]. Activation of the FimDCH complex was achieved by incubating it with either FimC–FimF or FimC–FimG. This activation step was conducted for 25 min at 23 °C in 20 mM Tris-HCl [pH 8.0], 50 mM NaCl and 0.05% n-dodecyl-ß-D-maltoside (DDM) resulting in a final concentration of 0.34 μM FimDCH and 2.8 μM FimC–FimF or FimC–FimG.

Pilus formation was initiated by the sequential addition of the FimA–FimC complex to the activated FimDCHF complex. Specifically, a total of 2 mL of 70 μM FimA–FimC complex in 20 mM Tris [pH 8.0], 50 mM NaCl and 0.05% DDM was added in four increments of 500 μL to the activated FimDCHF complex over 270 min, resulting in a final FimDCH concentration of 0.16 μM and a final FimA concentration of 40 μM (~250-fold molar excess of FimA over FimD). The progress of the reaction was followed by analytical cation exchange chromatography on a 1 mL Resource S column (Cytiva) at different time points via the disappearance of the FimA–FimC peak. Assembled pili (termed FimD-cat) were purified as described above.

### Production of isotope-labelled type 1 pilus rods

To produce uniformly $^{13}$C, $^{15}$N-labelled pili in vivo, cells were grown in M9 minimal media supplemented with $^{13}$C–glucose (4 g/L) (Sigma) and $^{15}$N–ammonium chloride (0.5 g/L) (Sigma). The 6 × 1.5 L culture was grown at 37 °C to an optical density (OD$_{600}$) of 0.8, induced with 0.1% arabinose and further incubated overnight. All other aspects of pilus expression and subsequent purification were performed as described above for the production of short pili from the Fim-Ara plasmid. For the in vitro production of uniformly $^{13}$C, $^{15}$N-labelled pili, FimA was produced as described[46], but with isotope-supplemented M9 media (as above). The purification of FimA and its subsequent assembly into pilus rods was carried out as described in ref. 46. For the production of $^{13}$C, $^{15}$N-labelled FimD-cat pili, FimA was produced as described above using isotope-supplemented M9 media. Labelled FimA was then refolded by rapid dilution in the presence of FimC to form FimA–FimC complexes and FimD-catalysed assembly was performed as described earlier for the preparation of unlabelled FimD-cat pili. All labelled pili were pelleted by ultracentrifugation at 200,000 × *g*, for 2 h at 4 °C with an MLA-80 rotor (Beckman Coulter) followed by washing with 50 mM Tris-HCl [pH 8.0], 1 mM EDTA. These steps were repeated four times before finally resuspending in 500 μL of the same buffer.

## Solid-state NMR data acquisition

All protein solutions were sedimented[61,62] in the MAS-NMR rotors (16 h at 4 °C at 210,000 × g) using home-built filling tools[63]. Solid-state NMR spectra were acquired on a wide-bore 850 MHz Bruker Avance III spectrometer employing a 3.2 mm Bruker Biospin 'E-free' probe[64]. The MAS frequency was set to 17.0 kHz and the sample temperature was adjusted to 278 K using the water line as an internal reference[63]. For all spectra, the $^1$H, $^{13}$C cross-polarisation (CP) step was achieved by using a $^1$H rf-field of 60 kHz and a $^{13}$C rf-field of 43 kHz and a contact time of 700 µs. The $^1$H, $^{15}$N CP step was optimised to 60 kHz rf-field on $^1$H and 43 kHz rf-field on $^{15}$N and a CP contact time of 1 ms. The $^{15}$N, $^{13}$C SPECIFIC CP step used in the 2D NCA experiments was achieved by using rf-fields of 6 kHz and 11 kHz on $^{13}$C and $^{15}$N, respectively. The CP length was optimised to 7 ms. For the 2D DARR spectra, 3072 points in the direct and 2560 points in the indirect dimensions were chosen (acquisition times of 15.4 and 12.8 ms, respectively). The repetition rate was set to 2.7 s and 8–16 number of scans were used. The 2D NCA spectra were recorded with 3072 and 1536 points in the direct and indirect dimensions leading to acquisition times of 15.4 and 11.5 ms, respectively. The repetition rate was set to 3 s and 8–12 number of scans were used. SPINAL-64 proton decoupling with an rf-field of 90 kHz was used in all experiments. The spectra were processed with the software TOPSPIN (version 3.5, Bruker Biospin) with a shifted (3.0) squared cosine apodization function and automated baseline correction in the indirect and direct dimension. Spectral analysis was performed using CcpNmr Analysis 2.4.2[65–67]. Spectra were referenced to 4,4-dimethyl-4-silapentane-1-sulfonic acid (DSS).

## Temperature-induced unfolding transitions

Pilus samples (400 µM FimA monomers) were diluted to a final FimA monomer concentration of 20 µM with $H_3PO_4$–NaOH [pH 7.0], 7.5 M GdmCl. The far-UV CD signal at 230 nm was monitored while heating the sample from 25–95 °C at a rate of 1 °C/min. Data were fitted with OriginPro 2017 according to a two-state van't Hoff transition (Eq. 1) and only the value for the melting temperature was extracted for analysis[68].

$$S = ((S_f + m_f * T) + (S_u + m_u * T) * \exp(\Delta H_m/(R * T) \\ * (T - T_m)/T_m))/(1 + \exp(\Delta H_m/(R * T) * (T - T_m)/T_m)) \tag{1}$$

where $S$ is the measured CD signal, $S_u$ and $S_f$ are the signals of unfolded and folded protein at zero K, $m_u$ and $m_f$ are the temperature dependencies ('slopes') of the signal of unfolded and native protein, respectively, $T_m$ is the melting temperature, $\Delta H$ is the enthalpy of unfolding, $R$ the gas constant and $T$ the temperature in K.

## Unfolding and dissociation kinetics

Different pilus preparations (400 µM) in $ddH_2O$ were diluted 1:40 (with manual mixing) in 100 mM $H_3PO_4$–NaOH [pH 2.1] containing different GdmCl concentrations and the kinetics of dissociation/unfolding was monitored by the change in the far-UV CD at 230 nm for type 1 pili or 222 nm for P pili, using a Jasco J-715 spectropolarimeter with 3 nm slits. Final GdmCl concentrations were verified via their refractive index[69]. All measured dissociation/unfolding traces were consistent with a single first-order reaction and fitted with the following equation,

$$S = S_0 + (S_\infty - S_0) * e^{-k_U t} \tag{2}$$

where $S_0$ is the signal at $t = 0$, $S_\infty$ is the signal at the end of the reaction, and $k_U$ is the first-order rate constant of unfolding. The natural logarithm of the rate constants was plotted against GdmCl concentration and fitted with a linear function (for in vitro-assembled pili) or with a model with a high-energy, on-pathway intermediate[30,47], in cases

when a deviation from linearity was observed (FimD-cat pili and in vivo pili) (Eq. 3),

$$N \underset{k_{IN}}{\overset{k_{NI}}{\rightleftarrows}} I^* \underset{k_{UI}}{\overset{k_{IU}}{\rightleftarrows}} U \tag{3}$$

where $N$ is the native state, $I^*$ is intermediate, $U$ is the unfolded state, and the $k$ parameters indicate the respective first-order rate constants. To fit the data, $k_{UI}$ was set to zero and exponential dependencies of the rate constants $k_{NI}$, $k_{IN}$ and $k_{IU}$ on denaturant concentration were used. The $\ln(k_U^{obs})$ dependence on the GdmCl concentration was then described by Eq. 4:

$$\ln\left(k_U^{obs}\right) = \ln\left(\frac{k_{NI} + k_{IN} + k_{IU} - \sqrt{(k_{NI} + k_{IN} + k_{IU})^2 - 4k_{NI}k_{IU}}}{2}\right) \tag{4}$$

## Protein concentrations

Concentrations of purified proteins were measured by their specific absorption at 280 nm, using the following molar extinction coefficients: FimA: 2680 $M^{-1}$ $cm^{-1}$, PapA: 3105 $M^{-1}$ $cm^{-1}$, PapD: 26,025 $M^{-1}$ $cm^{-1}$, PapAD: 29,130 $M^{-1}$ $cm^{-1}$, FimDCH: 202,000 $M^{-1}$ $cm^{-1}$; FimCF: 33,015 $M^{-1}$ $cm^{-1}$; FimCG: 36,000 $M^{-1}$ $cm^{-1}$; FimCA: 26,680 $M^{-1}$ $cm^{-1}$; FimC: 24,320 $M^{-1}$ $cm^{-1}$.

## Cryo-EM sample preparation and data collection

Type 1 pilus samples (in vitro-assembled, in vivo-assembled and FimD-cat) were diluted to 100 µM using $ddH_2O$. Quantifoil 1.2/1.3 400 mesh copper support grids were glow discharged for 45 s with a current of 25 mA. Vitrification was performed by applying 3 µL of pilus sample to the grid, incubating for 30 s, blotting for 6 s and plunging the grid into liquid ethane using a Vitrobot Mark IV at 4 °C and a humidity of ~70%. Data were collected in three separate sessions on FEI Titan Krios microscopes, at the UK national electron bio-imaging centre (eBIC) and ETH Zürich, operated at 300 kV and equipped with a K2 Summit direct electron detector (Gatan) and placed at the end of a GIF quantum energy filter (Gatan) with a slit width of 20 eV. A summary of acquisition parameters for each dataset is provided in Table S2.

## Cryo-EM image processing and reconstruction

All three datasets were processed in Relion-3.08[70] using the embedded helical reconstruction functions[71]. First, all movies were dose-weighted using Relion's implementation of MotionCorr. Next, contrast transfer function (CTF) parameters were estimated with Gctf 1.06[72]. The description of the reconstruction below refers to the in vivo-assembled type 1 pilus dataset, but all three structures were processed in an analogous manner. From the in vivo-assembled pilus dataset, the ~1400 best CTF-corrected micrographs were selected for further processing. Fifty filaments were manually picked, and segments were extracted with a box size of 256 px. The 2D classification was performed by skipping the first CTF peak and limiting the resolution to 5 Å. The best 2D classes were used as references for filaments auto-picking from the whole dataset. In total, ~530 k segments were extracted using the same box size of 256 px, and 2D classification was repeated using the same settings. The best-looking 2D classes were selected comprising a total of 160 k particles and used for 3D classification. 3D classification gave rise to a class containing 40 k particles displaying high-resolution features. The first 3D refinement of the best 3D class resulted in 3.81 Å resolution after post-processing in Relion. After the first round of particle polishing and per-particle CTF refinement, the resolution improved to 3.18 Å. Further rounds of particle polishing and CTF refinement resulted in a final resolution of 2.85 Å as assessed by the gold standard Fourier shell correlation (FSC) procedure (FSC = 0.143)[73].

## Model building, refinement and structure analysis

The previously reported structure of the in vivo-assembled type 1 pilus rod (PDB ID: 5OHO[28]) was used as a starting model for model building and refinement. First, the model was fitted into the cryo-EM map using Chimera[74], followed by several cycles of real space refinement in PHENIX[75]. Each refinement cycle was performed without additional restraints and was followed by iterative manual building in *Coot* 0.8.9[76]. All figures were produced in ChimeraX[77] and PyMOL 2.4 (Schrödinger, LLC).

## Calculating and analysis of contact maps

Atomic models were aligned globally using the *align* command in PyMol 2.4 (Schrödinger, LLC) and saved as separate files. A python script was used to calculate distances between all atoms in the structure interactively and to determine whether two atoms were within 8 Å distance of each other. Calculated contact maps were saved and compared with contact maps of other structures by simple addition. Contacts present in both structures resulted in a value of 2, and all different interactions were equal to 1. Next, all interactions were summed for each interface. For single model analysis, all the same interfaces were compared for all chains, and the number of different interfaces and total interfaces was plotted. Only a global comparison was made for comparing two structures without averaging the same interfaces between different subunits.

## Analytical size exclusion chromatography

All analytical size exclusion chromatography (SEC) experiments were performed at 25 °C using a Superose 6 increase 10/300 GL column (Cytiva) equilibrated with running buffer (20 mM sodium acetate [pH 5.0], 150 mM NaCl) on an HPLC instrument equipped with an auto sampler (Agilent). Eluted proteins were detected via their absorption at 230 nm. For each SEC experiment, 5 μL of pilus containing solution (100 μM FimA monomers) was mixed with 45 μL of denaturant solution at various GdmCl concentrations (0–7.5 M) in 100 mM $H_3PO_4$ [pH 2.0]. Following incubation at 25 °C for 19.5 or 25.5 h, samples were diluted twofold with non-denaturing running buffer to quench the dissociation/unfolding reaction and loaded onto the column for analysis. To analyse the kinetics of pilus fragmentation, pilus solutions were first mixed with concentrated denaturant solution to a final concentration of 4.1 M GdmCl. Samples were then taken at various time points (0, 5 and 60 min), diluted twofold with running buffer and loaded onto the column.

## In vitro pili fragmentation and reassembly

To analyse reassembly of fragmented pili, in vitro-assembled pili (380 μM) were fragmented in 4.5 M GdmCl for 5 min. The reaction was diluted twofold with SEC running buffer (20 mM sodium acetate [pH 5.0], 150 mM NaCl) to halt further fragmentation and then the sample was fully buffer exchanged into running buffer and reconcentrated to a final FimA monomer concentration of 200 μM. Concentrated fragments were incubated for 16 h at 37 °C to allow for pilus reassembly and then analysed by analytical SEC. The stability of the resulting pilus sample against GdmCl was measured as described above.

## Reporting summary

Further information on research design is available in the Nature Portfolio Reporting Summary linked to this article.

## Data availability

The electron microscopy (EM) maps reported in this paper have been deposited in the Electron Microscopy Data Bank (EMDB) with the following accession codes and descriptions: EMD-17863 (in vitro pili); EMD-10721 (in vivo pili); and EMD-17878 (FimD-catalysed pili). Correspondingly, the model coordinates have been deposited in the Protein Data Bank (PDB) under accession codes: 8PSV (in vitro pili); 6Y7S (in vivo pili); and 8PTU (FimD-catalysed pili). The raw data pertaining to all kinetic stability measurements can be found in the Source Data file published alongside this article. All other data supporting the findings of this study are available from the corresponding author upon request. Source data are provided with this paper.

## Code availability

The code for the Python script used in this study has been deposited in the GitHub repository [https://github.com/dzyla/pilus_stability].

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

## Acknowledgements

This work was funded by Swiss National Science Foundation grants 310030_201234, 310030B_176403/1 and 31003A_156304 to R.G., MRC grant 018434 to G.W. and Swiss National Science Foundation grant 200020_188711 to B.H.M. We would like to thank the Functional Genomics Center Zurich, especially Dr. Serge Chesnov for the help with mass spectrometric analyses of proteins. We wish to thank the UK National Electron Bio-Imaging Centre (eBIC) for access to cryo-EM facilities and data collection. We also wish to thank Dr. Natasha Lukoyanova, Miroslav Peterek and Dr. Daniel Böhringer for assistance with cryo-EM data acquisition. T.W. acknowledges support from the DFG Deutsche Forschungsgemeinschaft (Heisenberg fellowship, project number 455238107) and the Max Planck Society.

## Author contributions

D.S.Z. designed, measured and analysed all in vitro experiments, collected cryo-EM data for the in vivo sample, performed image processing, and built and refined all models. M.K.H. prepared cryo-EM grids for in vitro and FimD-cat pili, collected data and performed image processing. D.S.Z. and P.B. prepared pilus samples for solid-state NMR studies. R.Z. assisted with cryo-EM grid screening and data acquisition. T.W. acquired and analysed solid-state NMR data. C.G. designed the in vitro FimD-catalysed pilus assembly. G.W., R.G., M.K.H. and B.H.M. supervised the work. M.K.H. wrote the manuscript with help from all other authors.

## Competing interests

The authors declare no competing interests.
