## [Peer Review File · Nature Communications]

The assembly platform FimD is required to obtain the most stable quaternary structure of type 1 piliReviewer #1 (Remarks to the Author):

The manuscript by Zyla et al. investigates the basis for differences between type 1 pili assembled through an in vitro reaction, in the absence of the FimD usher assembly platform, versus native pili purified from the bacterial surface. Previous work from this group had demonstrated that in vitro-assembled type 1 pilus rods have ~1,000-fold lower stability compared to native pilus rods. In the current study, the authors investigated the basis for this difference using structural and biophysical approaches. Subtle structural differences between the in vitro and native type 1 rods were detected using NMR, but not cryo-EM. In vitro assembly in the presence of the usher was found to restore native stability to the pili and the authors propose that the usher controls pilus assembly to prevent the introduction of structural defects. The authors also show that P pilus rods exhibit similar behavior. Overall, this is a carefully done study with interesting implications regarding the function of the usher in optimal pilus assembly, although how to interpret the defects of the in vitro pili in terms of pilus assembly mechanism and relevance to biological assembly via the usher remain unclear. In addition, a potential role for the pilus tip subunits, which are added together with FimD in the in vitro reaction, should be considered. Specific comments about the manuscript are presented below.

Major points:

1. The FimD-cat in vitro pili were assembled not only in the presence of FimD, but FimD primed with pilus tip subunits. The presence and exact composition of the pilus tip subunits used in the reactions needs to be clearly described in the text, and a potential role for these subunits in the stability of the assembled rods should be considered. Fig. 2a suggests that FimD primed with complete tip complexes (FimC-FGH) were used in the in vitro reactions. However, these reactions were in fact done with FimDCH and the addition of FimCF, correct? Fig. S4 indicates an experiment was also done with FimDCH and FimCG. This needs to be clarified in the results and methods sections.
2. Is there a way to test if the pilus tip subunits, with or without FimD, are required to achieve the optimal pilus rod stability? What happens if only FimDCH (no FimCF or CG) is added into the reaction? Or if FimCF is added into the in vitro reaction in the absence of FimD? Understanding this seems critical to the conclusions of the paper.
3. The final set of experiments (p.9-10, Fig. 6) suggest that the lower stability of the in vitro-assembled rods may be due to defects arising from en bloc incorporation of smaller rod fragments to form longer rods. If this is the case, then longer lengths assembled in vitro, without incorporating smaller fragments, should be just as stable as the in vivo or usher-assembled pili. If the fraction corresponding to the void volume peak observed in Fig. 6b at 0 or 4.1 M GdmCl is isolated and then re-analyzed, does it now run as a single peak and have increased stability?
4. If the lower stability of the in vitro-assembled rods (in the absence of the usher) is due to defects from en bloc incorporation, could this be interpreted as an artifact allowed by the in vitro system, rather than suggesting a specialized function of the usher to ensure optimal pilus stability? The chaperone and usher both likely function in vivo to ensure linear and processive subunit incorporation. The chaperone functions to prevent premature subunit assembly in the periplasm, thereby preventing pilus fragments from forming, and the usher likely would likely reject any polymerized pilus fragments presented to it as these could not be accommodated by the usher's periplasmic assembly sites.

Minor points:

5. Line 55: Delete "PapE" (PapE is mentioned at the end of the sentence).
6. Line 55-56: The pilus rod also includes a terminator subunit (PapH for P pili, potentially FimI for type 1 pili). This comment also applies to Fig. S1a and the accompanying legend.
7. Line 72: Along with References 18-20, the recent paper showing P pilus assembly intermediates should also be cited (Du et al., <https://doi.org/10.1038/s41467-021-25522-6>).
8. Line 76/Reference 24: Is there a published reference regarding a role for FimI as a rod terminator? Reference 24 is a dissertation?
9. There are inconsistencies between the text and figures in terms of the labeling for the kinetic constants. Fig. 1b indicates a k_F , the legend refers to a k_{obs} , and the Results text (for example, line 130) refers to a k_U . Also, subsequent figures indicate a $t_{1/2}$ instead of k_F . Indication of $t_{1/2}$ in Fig 1

would facilitate comparison among the results.

Reviewer #2 (Remarks to the Author):

The manuscript describes an overall thorough investigation of type I pili, with a focus on the correlation between a) structural differences and b) differences in stability, using various pili preparations. Specifically, *in vitro* and *in vivo* pili are investigated. For the *in vitro* case, pili are compared that are produced i) from FimA alone ii) from FimA-FimC complexes, and iii) additionally in the presence of the usher FimD. Using various structural techniques the different resulting pili are investigated. Cryo-EM (from the present study) could not reveal any significant structural differences between *in vitro* and *in vivo* pili, despite very high resolution. Therefore the authors resorted to solid-state NMR and observed in high quality solid-state NMR spectra different degrees of polymorphism in their samples, where *in vivo* pili exhibit cleaner spectra and *in vitro* pili exhibit for some residues a second conformation, that vanishes when *in vitro* pili are prepared in the presence of FimD. The authors conclude - and I think correctly - that FimD is important for correct assembly of defect-free pili that are more stable than *in vitro* pili that were produced without FimD.

I believe the manuscript could certainly become interesting for Nature Communications but some parts should be improved before acceptance:

- i) The study relies heavily on solid-state NMR. One key argument is that *in vitro* samples consistently exhibit polymorphism and that in the hands of the authors two *in vitro* samples show the same intensity ratios for major/minor peaks. (Table S3 - not quite the same ratio for V17). However, this appears to depend also on the details of the sample preparation as in a previous solid-state NMR study of *in vitro* samples less polymorphism was observed. see e.g. T128 in the NCA spectrum as shown in ref [32] SI Fig 3.
- ii) Polymorphism in solid-state NMR spectra of *in vitro* fibrils has been observed before and described in the literature: see PMID: 31028572. These data need to be discussed.
- iii) Only CC and NCA 2D spectra are presented. The spectral quality seems really nice, why didn't the authors record a more extensive data set or even proton-detected data (as in PMID: 31028572)?
- iv) Is the polymorphism stronger in the side chains than in the back bone? For this a 3D NCACX could be useful. Table S3 currently refers only to single peaks per residue, and doesn't give a complete picture.

Reviewer #3 (Remarks to the Author):

The manuscript of Zyla et al. is focused on understanding the assembly and quaternary structure formation of type 1 pili. Authors used NMR, cryo-electron microscopy and circular dichroism to test 3 assembly conditions of type 1 pili: formed *in vivo*, *in vitro* (-FimD) and *in vitro* (+FimD). The main finding of the study is that the FimD usher assembly platform is required for the formation of the most stable quaternary structure of the pilus. Pili assembled *in vitro* without FimD harbour structural defects that compromise their stability. It is however unclear what these structural defects are. Unfortunately, the cryoEM maps of the pili did not reveal any significant differences between all three pili samples, and the NMR data showed minor chemical-shift differences and peak splitting for a subset of residues. Therefore, the reported structural defects were not identified, and the study lacks the mechanistic understanding of the instability of *in vitro* formed pili. Thus, the reviewer believes that a number of additional experiments must be performed before the work can be accepted for the publication in

Nature Communications. MD simulations can help to gain additional insights into the quaternary structure formation with and without FimD. Perhaps, in the absence of the usher there are some unfavourable contacts between pilin subunits that contribute to the lower stability. Furthermore, site-directed mutagenesis of the residues revealed by NMR might show whether these residues contribute to the instability of the in vitro pili.

Furthermore, the reviewer have several additional questions/comments:

Major comments:

1. The authors determined the structure of in vivo assembled type 1 pili to a higher resolution than published previously (from 4.2 Angstrom to 2.9 Angstrom). These two models need to be compared. Does the improvement in resolution provide any additional insights into the pilus structure or reveal any additional contacts?
2. For the in vitro assembly, the proteins were incubated at 37 degrees for 14 days. The reviewer is wondering if the structural defects observed in the quaternary structure are, at least in part, the result of the assembly conditions? The appropriate control would be to test the stability of the isolated in vivo pili after incubation at 37 degrees for 14 days.
3. The critical role of the usher assembly platform for quaternary structure formation has been previously reported for e.g. Csu pili from archaic chaperone-usher pathway (Pakharukova et., 2020). As this is the main finding of the manuscript in question, it is of utmost importance that the authors refer to the previously done studies on other pili systems and mention it in Discussion.
4. For Discussion: what is the biological significance, if any, of spontaneous pili assembly without the usher? Obviously the pilus cannot be secreted without FimD, would these polymers just accumulate in the periplasm?

Minor comments:

1. Table S1: please indicate the PDB and EMD codes for each model in the table.
2. Please correct the bond angle outliers in the model of the in vivo pilus.

Reviewer Comments

Reviewer 1:

The manuscript by Zyla et al. investigates the basis for differences between type 1 pili assembled through an *in vitro* reaction, in the absence of the FimD usher assembly platform, versus native pili purified from the bacterial surface. Previous work from this group had demonstrated that *in vitro*-assembled type 1 pilus rods have ~1,000-fold lower stability compared to native pilus rods. In the current study, the authors investigated the basis for this difference using structural and biophysical approaches. Subtle structural differences between the *in vitro* and native type 1 rods were detected using NMR, but not cryo-EM. *In vitro* assembly in the presence of the usher was found to restore native stability to the pili and the authors propose that the usher controls pilus assembly to prevent the introduction of structural defects. The authors also show that P pilus rods exhibit similar behavior. Overall, this is a carefully done study with interesting implications regarding the function of the usher in optimal pilus assembly, although how to interpret the defects of the *in vitro* pili in terms of pilus assembly mechanism and relevance to biological assembly via the usher remain unclear. In addition, a potential role for the pilus tip subunits, which are added together with FimD in the *in vitro* reaction, should be considered. Specific comments about the manuscript are presented below.

1. The FimD-catalysed *in vitro* pili were assembled not only in the presence of FimD, but FimD primed with pilus tip subunits. The presence and exact composition of the pilus tip subunits used in the reactions needs to be clearly described in the text, and a potential role for these subunits in the stability of the assembled rods should be considered. Fig. 2a suggests that FimD primed with complete tip complexes (FimC-FGH) were used in the *in vitro* reactions. However, these reactions were in fact done with FimDCH and the addition of FimCF, correct? Fig. S4 indicates an experiment was also done with FimDCH and FimCG. This needs to be clarified in the results and methods sections.

We thank the reviewer for these comments and for identifying the confusing description of how our samples were assembled. Firstly, we have improved our description of the pilus assembly reactions in the text. The FimDCH complex was expressed and purified as described in Phan 2011. Our most recent study (Giese et al, *Nat Commun* **14**, 7718 (2023). <https://doi.org/10.1038/s41467-023-43449-y>) showed that FimD-catalysed assembly can only be initiated if the FimDCH complex is primed with FimC-FimF or FimC-FimG. For pilus assembly, we tested both conditions (but not in combination due to the difficulty of estimating the fraction of rods that had incorporated both FimF and FimG). Activated complexes, FimDCHF or FimDCHG, were subsequently used for FimD-catalysed *in vitro* assembly by sequential addition of FimC-FimA. The reason for this approach was to minimise uncatalysed FimC-FimA pilus assembly (*in vitro* assembly).

Secondly, we have also provided our reasoning in the text (both in the relevant results section and the discussion) as to why we think the presence of the tip fibrillum is not responsible for the observed differences in stability against dissociation/unfolding. Briefly, we believe that our assays measure the overall stability of our pilus samples against dissociation and subsequent unfolding, with subunit dissociation likely being the rate-limiting step. Furthermore, we do not believe that this GdmCl- or

temperature-induced dissociation is initiated at the pilus ends, but rather affects all subunits simultaneously and equally. In addition, Fig. S4 shows that the stability of FimD-cat pili is the same when FimA assembly is initiated with FimD bearing FimG or FimF as the last subunit of the tip fibrillum. This also speaks against the tip fibrillum's influence on FimD-cat pili's stability. In Figure 2a and legend to this figure, we now provide a more precise description of the FimD-catalysed assembly reactions in which either FimF or FimG was the last subunit of the tip fibrillum.

2. Is there a way to test if the pilus tip subunits, with or without FimD, are required to achieve the optimal pilus rod stability? What happens if only FimDCH (no FimCF or CG) is added into the reaction? Or if FimCF is added into the *in vitro* reaction in the absence of FimD? Understanding this seems critical to the conclusions of the paper.

We thank the reviewer for bringing this issue to light. No pilus assembly can be achieved if only FimDCH and FimA-FimC are present in the reaction. For successful pilus assembly, FimF or FimG are required to prime FimD. We have thus clarified our text to avoid this confusion. This was investigated in detail in a recent publication (Giese et al., *Nat Commun* **14**, 7718 (2023). <https://doi.org/10.1038/s41467-023-43449-y>) to which we now refer in the manuscript.

3. The final set of experiments (p.9-10, Fig. 6) suggest that the lower stability of the *in vitro*-assembled rods may be due to defects arising from *en bloc* incorporation of smaller rod fragments to form longer rods. If this is the case, then longer lengths assembled *in vitro*, without incorporating smaller fragments, should be just as stable as the *in vivo* or usher-assembled pili. If the fraction corresponding to the void volume peak observed in Fig. 6b at 0 or 4.1 M GdmCl is isolated and then re-analyzed, does it now run as a single peak and have increased stability?

We thank the reviewer for this question. The GdmCl-induced fragmentation at 4.1 M GdmCl of *in vitro*-assembled pilus rods is time-sensitive and eventually would proceed until all pilus rods dissociated to smaller fragments. This is demonstrated in Fig. S14d, where only smaller pilus fragments remained after incubation of *in vitro*-assembled pili at 4.5 M GdmCl for 60 min, and the void volume peak (fragments longer than ca. 250 nm) essentially had disappeared (for comparison, see Fig. 1b, showing that no unfolding of *in vitro* assembled pili can be observed at 4.5 M). Unfortunately, it is extremely difficult to design an experiment to compare *in vivo*-assembled and *in vitro*-assembled pilus rods of the exact same length, nor is it possible to make longer "defect-free" *in vitro*-assembled pilus rods.

4. If the lower stability of the *in vitro*-assembled rods (in the absence of the usher) is due to defects from *en bloc* incorporation, could this be interpreted as an artifact allowed by the *in vitro* system, rather than suggesting a specialized function of the usher to ensure optimal pilus stability? The chaperone and usher both likely function *in vivo* to ensure linear and processive subunit incorporation. The chaperone functions to prevent premature subunit assembly in the periplasm, thereby preventing pilus fragments from forming, and the usher likely would likely reject any polymerized pilus fragments presented to it as these could not be accommodated by the usher's periplasmic assembly sites.

We completely agree with the reviewer that the chaperone and usher function *in vivo* to ensure linear and processive subunit incorporation and that short rod fragments would not be assembled in the periplasm. One could indeed view this as an *in vitro* artifact, but we believe this is all a matter of perspective, as this by no means excludes that the usher indeed guides the formation of the most stable pilus rod structure by ensuring linear/processive subunit incorporation and perhaps also ensuring correct subunit positioning. In addition, if we had not investigated pilus stability *in vitro*, we would not have discovered that the usher leads to the formation of such exquisitely stable pilus rods *in vivo*.

We have slightly modified the text in the discussion to include the wording "linear and processive" and a sentence about potential subunit positioning.

Minor points:

5. Line 55: Delete "PapE" (PapE is mentioned at the end of the sentence).

Thank you. We have made this change.

6. Line 55-56: The pilus rod also includes a terminator subunit (PapH for P pili, potentially FimI for type 1 pili). This comment also applies to Fig. S1a and the accompanying legend.

We have made the suggested additions to the text and the figure.

7. Line 72: Along with References 18-20, the recent paper showing P pilus assembly intermediates should also be cited (Du et al., <https://doi.org/10.1038/s41467-021-25522-6>).

We apologise, this was an oversight on our part. We have now included this reference (Du et al., 2021) and also moved Omattage et al., 2018 to this sentence.

8. Line 76/Reference 24: Is there a published reference regarding a role for FimI as a rod terminator? Reference 24 is a dissertation?

We have replaced the citation of the PhD thesis with that of the original paper which just appeared in *Nat Comm*: Giese et al, *Nat Commun* **14**, 7718 (2023). <https://doi.org/10.1038/s41467-023-43449-y>

9. There are inconsistencies between the text and figures in terms of the labeling for the kinetic constants. Fig. 1b indicates a k_F , the legend refers to a k_{obs} , and the Results text (for example, line 130) refers to a k_U . Also, subsequent figures indicate a $t_{1/2}$ instead of k_F . Indication of $t_{1/2}$ in Fig 1 would facilitate comparison among the results.

We thank the reviewer for highlighting these inconsistencies. These have now been corrected (all in text to k_U^0 (k_F was a typing error), and Figure 1 has both, k_U at 6.45 M GdmCl ($k_U^{6.45M}$) that was used for

comparison as well as the reaction half-life. For clarity, we also added a new table (Table S1) that contains all fitting information for all pilus samples analysed in this manuscript. This addition should facilitate easier comparison between samples.

Reviewer 2:

The manuscript describes an overall thorough investigation of type I pili, with a focus on the correlation between a) structural differences and b) differences in stability, using various pili preparations. Specifically, *in vitro* and *in vivo* pili are investigated. For the *in vitro* case, pili are compared that are produced i) from FimA alone ii) from FimA-FimC complexes, and iii) additionally in the presence of the usher FimD. Using various structural techniques the different resulting pili are investigated. Cryo-EM (from the present study) could not reveal any significant structural differences between *in vitro* and *in vivo* pili, despite very high resolution. Therefore the authors resorted to solid-state NMR and observed in high quality solid-state NMR spectra different degrees of polymorphism in their samples, where *in vivo* pili exhibit cleaner spectra and *in vitro* pili exhibit for some residues a second conformation, that vanishes when *in vitro* pili are prepared in the presence of FimD. The authors conclude - and I think correctly - that FimD is important for correct assembly of defect-free pili that are more stable than *in vitro* pili that were produced without FimD.

I believe the manuscript could certainly become interesting for Nature Communications but some parts should be improved before acceptance:

i) The study relies heavily on solid-state NMR. One key argument is that *in vitro* samples consistently exhibit polymorphism and that in the hands of the authors two *in vitro* samples show the same intensity ratios for major/minor peaks. (Table S3 - not quite the same ratio for V17). However, this appears to depend also on the details of the sample preparation as in a previous solid-state NMR study of *in vitro* samples less polymorphism was observed. see e.g. T128 in the NCA spectrum as shown in ref [32] SI Fig 3.

We thank the reviewer for their comment. One important difference to the previous study mentioned, is that the type I pilus samples were not prepared in the same way. Our *in vitro* pili were assembled from FimA subunits that were incubated in 100 mM sodium acetate, pH 5.0, for 2 weeks at 37°C. This allowed for high-yield *in vitro* pilus assembly. As we tested in our stability assays, there was no difference between FimC-FimA and FimA-only *in vitro* assembled samples. Therefore, we did not proceed with FimC-FimA assembly of *in vitro* pili as described by Habenstein et al., mostly due to higher material needs, longer total incubation, and higher unspecific aggregation in our hands. We have clarified the differences in sample preparations to previous studies in the manuscript.

We agree with the reviewer that polymorphism of type 1 pili assembled *in vitro* in the absence of FimD in general, depends on the sample preparation. This is in line with the findings of Lange et al., where polymorphism has been observed in the samples used for ¹H-detected NMR experiments (Habenstein et al., *J. Biomol. NMR* **2019**, *73*, 281–291, very pronounced chemical-shift differences between the different polymorphs have been observed), but not in the ones used for ¹³C-detected NMR (Habenstein et al.,

Angew. Chem. Int. Ed. **2015**, *54*, 11691–11695). However, due to the reproducibility of the spectra for two independent preparations, we assume that in our case, polymorphism is most likely not responsible for the observed peak splittings, but instead, structural defects lead to the observation of peak doubling.

ii) Polymorphism in solid-state NMR spectra of in vitro fibrils has been observed before and described in the literature: see PMID: 31028572. These data need to be discussed.

We have mentioned the previous finding of polymorphism in FimA rods in the discussion section of our manuscript and have now added the appropriate citation.

iii) Only CC and NCA 2D spectra are presented. The spectral quality seems really nice, why didn't the authors record a more extensive data set or even proton-detected data (as in PMID: 31028572)?

While we agree with the reviewer that it is always nice to have more spectra, we think that the reported 2D spectra are sufficient to monitor the structural differences between the samples studied. Our goal with the NMR studies was not to determine the structure of the pilus rod (this we had already done by cryo-EM), but rather to pinpoint the structural defects responsible for our observed stability differences. Therefore, we reasoned that it was sufficient to perform a targeted set of experiments in order to pinpoint these and answer our question.

iv) Is the polymorphism stronger in the side chains than in the back bone? For this a 3D NCACX could be useful. Table S3 currently refers only to single peaks per residue, and doesn't give a complete picture.

If by "stronger polymorphism" the reviewer means bigger chemical-shift differences, then this is hard to judge for our data because the shift differences are small. Importantly, however, we do not support the polymorphic model (i) but a model (ii) in which packing errors appear along the pilus rods which then lead to a slightly different conformation and a chemical-shift difference. This differs from a polymorphism caused by a mixture of pili with homogeneous but slightly different quaternary structures. This is already stated in line 362 of the original manuscript (lines 389-392 of the revised version).

Reviewer 3:

The manuscript of Zyla et al. is focused on understanding the assembly and quaternary structure formation of type 1 pili. Authors used NMR, cryo-electron microscopy and circular dichroism to test 3 assembly conditions of type 1 pili: formed in vivo, in vitro (-FimD) and in vitro (+FimD). The main finding of the study is that the FimD usher assembly platform is required for the formation of the most stable quaternary structure of the pilus. Pili assembled in vitro without FimD harbour structural defects that compromise their stability. It is however unclear what these structural defects are. Unfortunately, the cryoEM maps of the pili did not reveal any significant differences between all three pili samples, and the NMR data showed minor chemical-shift differences and peak splitting for a subset of residues. Therefore, the reported structural defects were not identified, and the study lacks the mechanistic understanding of the instability of in vitro formed pili. Thus, the reviewer believes that a number of additional experiments must be performed before the work can be accepted for the publication in Nature Communications.

MD simulations can help to gain additional insights into the quaternary structure formation with and without FimD. Perhaps, in the absence of the usher there are some unfavourable contacts between pilin subunits that contribute to the lower stability.

We thank the reviewer for this suggestion; however, we do not believe that MD simulations would provide additional insights on top of those we have already gleaned in our experimental approaches. We are confident that our NMR studies have revealed the unfavorable contacts in our *in vitro*-assembled pilus rods (please also refer to our responses to Reviewer 2 above).

Furthermore, site-directed mutagenesis of the residues revealed by NMR might show whether these residues contribute to the instability of the *in vitro* pili.

Mutagenesis of residues in FimA (or PapA) is fraught with some important challenges. Although we agree that mutagenesis of certain residues within FimA would lead to the destabilization of the resulting pilus rod, these effects would almost certainly be different from the observed differences in stability against dissociation/ unfolding we observe in our experiments, which we propose are caused by structural packing defects. It would be almost impossible to choose an appropriate residue to mutagenise to, which would mimic what we see in our NMR data. Moreover, mutagenesis of FimA residues would likely lead to pilus rod assembly defects (as were observed for the mutagenesis of PapA, Hospenthal et al., 2016). Furthermore, we believe that mutagenesis of FimA would unnecessarily dilute the main conclusion of our study, namely that type 1 pilus rods only attain the most stable quaternary structure, free of defects, when their assembly is catalyzed unidirectionally by FimD.

Furthermore, the reviewer have several additional questions/comments:

1. The authors determined the structure of *in vivo* assembled type 1 pili to a higher resolution than published previously (from 4.2 Angstrom to 2.9 Angstrom). These two models need to be compared. Does the improvement in resolution provide any additional insights into the pilus structure or reveal any additional contacts?

Our new high-resolution structure, although very nice, did not reveal any major new insights that were not evident from the previous structures. This is why we have not included a direct comparison of these structures in the original manuscript. Nevertheless, we followed the reviewer's suggestion and now added a new supplementary figure (Fig. S7) to provide some comparison.

2. For the *in vitro* assembly, the proteins were incubated at 37 degrees for 14 days. The reviewer is wondering if the structural defects observed in the quaternary structure are, at least in part, the result of the assembly conditions? The appropriate control would be to test the stability of the isolated *in vivo* pili after incubation at 37 degrees for 14 days.

We thank the reviewer this comment. Our *in vivo*-assembled pili were produced by culturing bacteria at 37 degrees Celsius, and our *in vitro*-assembled pili were generated in various reactions at different

temperatures (Fig. S4). Therefore, we are confident that temperature does not have an impact on the quaternary structure heterogeneity/defects and, thus, the stability of pilus rods against dissociation/unfolding.

3. The critical role of the usher assembly platform for quaternary structure formation has been previously reported for e.g. Csu pili from archaic chaperone-usher pathway (Pakharukova et., 2020). As this is the main finding of the manuscript in question, it is of utmost importance that the authors refer to the previously done studies on other pili systems and mention it in discussion.

We apologize and have now included mention of this study in the discussion.

4. For discussion: what is the biological significance, if any, of spontaneous pili assembly without the usher? Obviously the pilus cannot be secreted without FimD, would these polymers just accumulate in the periplasm?

Please also refer to our response to Reviewers 1 and 2 above. We do not believe that rod segments or polymers would be assembled from FimC-FimA complexes in the periplasm, because the uncatalysed reaction is extremely slow (the second-order rate constant of formation of a FimA-FimA contact starting from FimA-FimC complexes is only $4 \text{ M}^{-1}\text{s}^{-1}$; see Vetsch et al., EMBO rep. 2006, <https://doi.org/10.1038/sj.embor.7400722>). The discussion already included "the in vivo assembly mechanism of CU pili does not allow spontaneous rod fragment assembly," but we have now modified this paragraph further for additional clarity.

Minor comments:

1. Table S1: please indicate the PDB and EMD codes for each model in the table.

Thank you, we have done this.

2. Please correct the bond angle outliers in the model of the in vivo pilus.

We have done this, and the PDB model was updated accordingly.

Reviewer #1 (Remarks to the Author):

The authors have addressed my concerns and the revised manuscript is appropriately updated and improved. I do not have additional concerns and support acceptance of this interesting study.

Reviewer #2 (Remarks to the Author):

The revised version of the paper is ready for acceptance.

Reviewer #3 (Remarks to the Author):

The authors have answered the reviewers questions and provided additional explanations in the manuscript. I am supportive of publication.